# FIFA: Making Fairness More Generalizable in Classifiers Trained on Imbalanced Data

**Zhun Deng** [*]
Columbia University
zd2322@columbia.edu

**Jiayao Zhang**
University of Pennsylvania
zjiayao@upenn.edu

**Linjun Zhang**
Rutgers University
lz412@stat.rutgers.edu

**Ting Ye**
University of Washington
tingye1@uw.edu

**Yates Coley**
Kaiser Permanente Washington
Health Research Institute
& University of Washington
Rebecca.Y.Coley@kp.org

**Weijie J. Su**
University of Pennsylvania
suw@wharton.upenn.edu

**James Zou**
Stanford University
jamesz@stanford.edu

## Abstract

Algorithmic fairness plays an important role in machine learning and imposing fairness constraints during learning is a common approach. However, many datasets are imbalanced in certain label classes (e.g. "healthy") and sensitive subgroups (e.g. "older patients"). Empirically, this imbalance leads to a lack of generalizability not only of classification, but also of fairness properties, especially in over-parameterized models. For example, fairness-aware training may ensure equalized odds (EO) on the training data, but EO is far from being satisfied on new users. In this paper, we propose a theoretically-principled, yet **F**lexible approach that is **I**mbalance-**F**airness-**A**ware (**FIFA**). Specifically, FIFA encourages both classification and fairness generalization and can be flexibly combined with many existing fair learning methods with logits-based losses. While our main focus is on EO, FIFA can be directly applied to achieve equalized opportunity (EqOpt); and under certain conditions, it can also be applied to other fairness notions. We demonstrate the power of FIFA by combining it with a popular fair classification algorithm, and the resulting algorithm achieves significantly better fairness generalization on several real-world datasets.

## 1 Introduction

Machine learning systems are becoming increasingly vital in our daily lives. The growing concern that they may inadvertently discriminate against minorities and other protected groups when identifying or allocating resources has attracted numerous attention from various communities. While significant efforts have been devoted in understanding and correcting biases in classical models such as logistic regressions and supported vector machines (SVM), see, e.g., (Agarwal et al., 2018; Hardt et al., 2016), those derived tools are far less effective on modern over-parameterized models such as neural networks (NN). Furthermore, in large models, it is also difficult for *measures of fairness* (such as equalized odds to be introduced shortly) to generalize, as shown in Fig. 1. In other

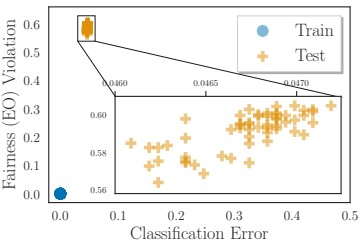

**Figure 1:** Each marker corresponds to a sufficiently well-trained ResNet-10 model trained on an imbalanced image classification dataset CelebA ((Liu et al., 2015)). The generalization of fairness constraints (EqualizedOdds) is substantially worse than the generalization of classification error.

---

[*]This work was done when the author was at Harvard University.

words, fairness-aware training (for instance, by imposing fairness constraints in training) may ensure measures of fairness on the training data, but those measures of fairness are far from being satisfied on test data. Here we find that sufficiently trained ResNet-10 models generalize well on classification error but poorly on fairness constraints—the gap in equalized odds between the test and training data is more than ten times larger than the gap for classification error between test and training.

In parallel, another outstanding challenge for generalization with real-world datasets is that they are often *imbalanced* across label and demographic groups (see Fig. 2 for imbalance in three commonly used datasets across various domains). This inherent nature of real-world data, greatly hinders the generalization of classifiers that are unaware of this innate imbalance, especially when the performance measure places substantial emphasis on minority classes or subgroups without sufficient samples (e.g., when considering the average classification error for each label class). Although generalizations with imbalanced data has been extensively studied and mitigation strategies are proposed (Cao et al., 2019; Mani & Zhang, 2003; He & Garcia, 2009; An et al., 2021; He & Ma, 2013; Krawczyk, 2016), it's unclear how well fairness properties generalize. And in this paper, we initiate the study of the open challenge: *how to ensure fairness generalization of over-parameterized models for supervised classification tasks on imbalanced datasets?*

**Our contributions.** Inspired by recent works on regularizing the minority classes more strongly than the frequent classes by imposing class-dependent margins (Cao et al., 2019) in standard supervised learning, we design a theoretically-principled, **F**lexible and **I**mbalance-**F**airness-**A**ware (FIFA) approach that takes both classification error and fairness constraints violation into account when training the model. Our proposed method FIFA can be flexibly combined with many fair learning methods with logits-based losses such as the soft margin loss (Liu et al., 2016) by encouraging larger margins for minority subgroups. While our method appears to be motivated for over-parameterized models such as neural networks, it nonetheless

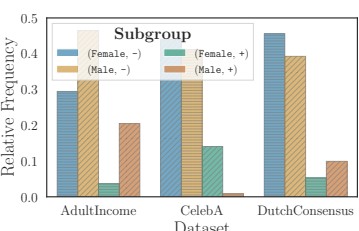

**Figure 2:** The subgroups (sensitive attribute, either `Male` or `Female`, and label class, either + or −) are very imbalanced in many popular datasets across different domains.

also helps simpler models such as logistic regressions. Experiments on both large datasets using over-parameterized models as well as smaller datasets using simpler models demonstrate the effectiveness, and flexibility of our approach in ensuring a better fairness generalization while preserving good classification generalization.

**Related work.** Supervised learning with imbalanced datasets have attracted significant interest in the machine learning communities, where several methods including resampling, reweighting, and data augmentation have been developed and deployed in practice (Mani & Zhang, 2003; He & Garcia, 2009; An et al., 2021). Theoretical analyses of those methods include margin-based approaches (Li et al., 2002; Kakade et al., 2008; Khan et al., 2019; Cao et al., 2019). Somewhat tangentially, an outstanding and emerging problem faced by modern models with real-world data is *algorithmic fairness* (Dwork et al., 2012; Coley et al., 2021; Deng et al., 2023), where practical algorithms are developed for pre-processing (Feldman et al., 2015), in-processing (Zemel et al., 2013; Edwards & Storkey, 2015; Zafar et al., 2017; Donini et al., 2018; Madras et al., 2018; Martinez et al., 2020; Lahoti et al., 2020; Deng et al., 2020), and post-processing (Hardt et al., 2016; Kim et al., 2019) steps. Nonetheless, there are several challenges when applying fairness algorithms in practice (Beutel et al., 2019; Saha et al., 2020; Deng et al., 2022; Holstein et al., 2019). Specifically, as hinted in Fig. 1, the fairness generalization guarantee, especially in over-parameterized models and large datasets, is not well-understood, leading to various practical concerns. We remark that although Kini et al. (2021) claims it is necessary to use multiplicative instead of additive logits adjustments, their motivating example is different from ours and they studied SVM with *fixed and specified budgets* for all inputs. Cotter et al. (2019) investigate the generalization of optimization with data-dependent constraints, but they do not address the inherent imbalance in real datasets, and their experimental results are not implemented with large neural networks used in practice. To the best of our knowledge, this paper is the first tackling the open challenge of fairness generalization with imbalanced data.

## 2 BACKGROUND

**Notation.** For any $k \in \mathbb{N}^+$, we use $[k]$ to denote the set $\{1, 2, \cdots, k\}$. For a vector $v$, let $v_i$ be the $i$-th coordinate of $v$. We use $\mathbf{1}$ to denote the indicator function. For a set $S$, we use $|S|$ to denote

the cardinality of $S$. For two positive sequences $\{a_k\}$ and $\{b_k\}$, we write $a_k = O(b_k)$ (or $a_n \lesssim b_n$), and $a_k = o(b_k)$, if $\lim_{k \to \infty} (a_k/b_k) < \infty$ and $\lim_{k \to \infty} (a_k/b_k) = 0$, respectively. We use $\mathbb{P}$ for probability and $\mathbb{E}$ for expectation, and we use $\hat{\mathbb{P}}$ and $\hat{\mathbb{E}}$ for empirical probability and expectation. For the two distributions $\mathcal{D}_1$ and $\mathcal{D}_2$, we use $p\mathcal{D}_1 + (1-p)\mathcal{D}_2$ for $p \in (0,1)$ to denote the mixture distribution such that a sample is drawn with probabilities $p$ and $(1-p)$ from $\mathcal{D}_1$ and $\mathcal{D}_2$ respectively. We use $\mathcal{N}_p(\mu, \Sigma)$ to denote $p$-dimensional Gaussian distribution with mean $\mu$ and variance $\Sigma$.

**Fairness notions.** Throughout the paper, we consider datasets consisting of triplets of the form $(x, y, a)$, where $x \in \mathcal{X}$ is a feature vector, $a \in \mathcal{A}$ is a sensitive attribute such as race and gender, and $y \in \mathcal{Y}$ is the corresponding label. The underlying random triplets corresponding to $(x, y, a)$ is denoted as $(X, Y, A)$. Our goal is to learn a predictor $h \in \mathcal{H} : \mathcal{X} \mapsto \mathcal{Y}$, where $h(X)$ is a prediction of the label $Y$ of input $X$. In this paper, we mainly consider *equalized odds* (EO) (Hardt et al., 2016) that has been widely used in previous literature on fairness. But our method could also be directly used to *equalized opportunity* (EqOpt) given that EqOpt is quite similar to EO. In addition, under certain conditions, our method could also be used to demographic parity (DP), which we will mainly discuss in the Appendix.

(i). *Equalized odds (EO) and Equalized opportunity (EqOpt).* A predictor $h$ satisfies equalized odds if $h(X)$ is conditionally independent of the sensitive attribute $A$ given $Y$: $\mathbb{P}(h(X) = y|Y = y, A = a) = \mathbb{P}(h(X) = y|Y = y)$. If $\mathcal{Y} = \{0, 1\}$ and we only require $\mathbb{P}(h(X) = 1|Y = 1, A = a) = \mathbb{P}(h(X) = 1|Y = 1)$, we say $h$ satisfies equalized opportunity.

(ii). *Demographic parity (DP).* A predictor $h$ satisfies demographic parity if $h(X)$ is statistically independent of the sensitive attribute $A$: $\mathbb{P}(h(X) = Y|A = a) = \mathbb{P}(h(X) = Y)$.

## 3  THEORY-INSPIRED DERIVATION

While we will formally introduce our new approach in Section 4, this section gives an informal derivation, with an emphasis on insights. We design an imbalance-fairness-aware approach that can be flexibly combined with fair learning methods with logits-based losses.

Throughout the paper, we use the lower letters, e.g. $x$, for realizations and capital letters, e.g. $X$, for random variables. Consider the supervised $k$-class classification problem, where a model $f : \mathcal{X} \mapsto \mathbb{R}^k$ provides $k$ scores, and the label is assigned as the class label with the highest score. The corresponding predictor $h(x) = \arg\max_i f(x)_i$ if there are no ties. Let us use $\mathcal{P}_i = \mathcal{P}(X|Y = i)$ to denote the conditional distribution when the class label is $i$ for $i \in [k]$ and $\mathcal{P}_{\text{bal}}$ to denote the balanced distribution $\mathcal{P}_{\text{Idx}}$, where Idx is uniformly drawn from $[k]$, i.e. $\sum_{i=1}^k \mathcal{P}_i/k$. $\mathcal{P}_{\text{bal}}$ can be viewed as a distribution weighting each class equally. Similarly, let us use $\mathcal{P}_{i,s} = \mathcal{P}(X|Y = i, A = s)$ to denote the conditional distribution when $Y = i$ and $A = s$. The corresponding empirical distributions induced by the training data are $\hat{\mathcal{P}}_i$, $\hat{\mathcal{P}}_{\text{bal}}$ and $\hat{\mathcal{P}}_{i,s}$. For the training dataset $\{(x_j, y_j, a_j)\}_j$, let $S_i = \{j : y_j = i\}$, $S_{i,a} = \{j : y_j = i, a_j = a\}$, and the corresponding sample sizes be $n_i$ and $n_{i,a}$, respectively. Although $\mathcal{P}_i$, $\mathcal{P}_{\text{bal}}$ and $\mathcal{P}_{i,s}$ are all distributions on $\mathcal{X}$, we sometimes use notations like $(X, Y) \sim \mathcal{P}_i$ to denote the distribution of $(X, i)$, where $X \sim \mathcal{P}_i$. In classical imbalanced data analysis, the goal is to ensure a small $\mathcal{L}_{\text{bal}}[f] = \mathbb{P}_{(X,Y) \sim \mathcal{P}_{\text{bal}}}[f(X)_Y < \max_{l \neq Y} f(X)_l]$. For our goal, we **not only** want to ensure a small $\mathcal{L}_{\text{bal}}[f]$, we **also** hope to keep the fairness violation error to be as small as possible. In order to do that, we need to take the margin of subgroups divided according to sensitive attributes in each label class (so called demographic subgroups in different classes) into account.

**Margin trade-off between classes of equalized odds.** In the setting of standard classification with imbalanced training datasets such as in Cao et al. (2019); Sagawa et al. (2020), the aim is to reach a small balanced test error $\mathcal{L}_{\text{bal}}[f]$. However, in a fair classification setting, our aim is not only to reach a small $\mathcal{L}_{\text{bal}}[f]$, but also to satisfy certain fairness constraints at *test time*. Specifically, for EO, the aim is:

$$\min_f \mathcal{L}_{\text{bal}}[f]$$

$$\text{s.t. } \forall y \in \mathcal{Y}, a \in \mathcal{A}, \ \mathbb{P}(h(X) = y|Y = y, A = a) = \mathbb{P}(h(X) = y|Y = y),$$

where we recall that $h(\cdot) = \arg\max_i f(\cdot)_i$. We remark here that in addition to the class-balanced loss $\mathcal{L}_{\text{bal}}[f]$, we can also consider the loss function that is balanced across all demographic subgroups in different classes, the derivation is similar and we omit it here.

Recall our motivating example in Figure 1. Whether the fairness violation error is small *at test time* should also be taken into account. Thus, our ***performance criterion*** for optimization should be:

$$\mathcal{L}_{\text{bal}}[f] + \alpha \mathcal{L}_{\text{fv}}, \tag{1}$$

where $\mathcal{L}_{\text{fv}}$ is a measure of fairness constraints violation that we will specify later, and $\alpha$ is a weight parameter chosen according to how much we care about the fairness constraints violation.

For simplicity, we start with $\mathcal{Y} = \{0, 1\}$ and $\mathcal{A} = \{a_1, a_2\}$. In the Appendix, we will further discuss the case when there are *multiple classes and multiple demographic groups*. We also want to clarify here the case we study is different from by naively viewing each demographic groups as a class and applying the method in Cao et al. (2019). The main difference is that our aim is to identify the class labels at test time, and we do not assume we have access to sensitive attributes at test time. As a result, the method in Cao et al. (2019) can not be directly used. Given that we mainly consider over-parameterized models, we assume the training data is well-separated that all the training samples are perfectly classified and fairness constraints are perfectly satisfied. The setting has been considered in (Cao et al., 2019) and can be satisfied if the model class is rich, for instance, for over-parameterized models such as neural networks. We also want to emphasize even though our theory-derived method assumes well-separation, our method can be applied to not well-separated datasets, please refer to the Section 6 for more details. If all the training samples are classified perfectly by $h$, not only $\mathbb{P}_{(X,Y)\sim\hat{\mathcal{P}}_{\text{bal}}}(h(X) \neq Y) = 0$ is satisfied, we also have that $\mathbb{P}_{(X,Y)\sim\hat{\mathcal{P}}_{i,a_j}}(h(X) \neq Y) = 0$ for all $i \in \mathcal{Y}$ and $a_j \in \mathcal{A}$. We remark here that $\mathbb{P}(h(X) = i | Y = i, A = a) = 1 - \mathbb{P}_{(X,Y)\sim\mathcal{P}_{i,a}}(h(X) \neq Y)$. Our performance criterion for optimization in (A.4) is:

$$\mathcal{M}[f] = \mathcal{L}_{\text{bal}}[f] + \alpha \sum_{i\in\mathcal{Y}} |\mathbb{P}(h(X) = i | Y = i, A = a_1) - \mathbb{P}(h(X) = i | Y = i, A = a_2)|.$$

By using classical margin theory bounds, We can establish the connection between margins for each demographic subgroups and the generalization performance in classification as well as fairness constraint, as proved in Theorem 3.1. Denote the margin for class $i$ by $\gamma_i = \min_{j\in S_i} \gamma(x_j, y_j)$, where $\gamma(x, y) = f(x)_y - \max_{l\neq y} f(x)_l$. One natural way to choose $\mathcal{L}_{\text{fv}}$ is to take $\sum_{i\in\mathcal{Y}} |\mathbb{P}(h(X) = i | Y = i, A = a_1) - \mathbb{P}(h(X) = i | Y = i, A = a_2)|$.

**Theorem 3.1 (Informal)** *With high probability over the randomness of the training data, for $\mathcal{Y} = \{0, 1\}$, $\mathcal{A} = \{a_1, a_2\}$, and for some proper complexity measure of class $\mathcal{F}$, i.e. $C(\mathcal{F})$ (see more details in the Appendix), for any $f \in \mathcal{F}$,*

$$\mathcal{M}[f] \lesssim \sum_{i\in\mathcal{Y}} \frac{1}{\gamma_i} \sqrt{\frac{C(\mathcal{F})}{n_i}} + \sum_{i\in\mathcal{Y},a\in\mathcal{A}} \frac{2\alpha}{\gamma_{i,a}} \sqrt{\frac{C(\mathcal{F})}{n_{i,a}}} \leq \sum_{i\in\mathcal{Y}} \frac{1}{\gamma_i} \sqrt{\frac{C(\mathcal{F})}{n_i}} + \sum_{i\in\mathcal{Y},a\in\mathcal{A}} \frac{2\alpha}{\gamma_i} \sqrt{\frac{C(\mathcal{F})}{n_{i,a}}},$$

$$\tag{2}$$

*where $\gamma_i$ is the margin of the $i$-th class's sample set $S_i$ and $\gamma_{i,a}$ is the margin of demographic subgroup's sample set $S_{i,a}$.*

Optimizing the upper bound in (2) with respect to margins in the sense that $g(\gamma_0, \gamma_1) \leq g(\gamma_0 - \delta, \gamma_1 + \delta)$ for $g(\gamma_0, \gamma_1) = \sum_{i\in\mathcal{Y}} \frac{1}{\gamma_i\sqrt{n_i}} + 2\alpha \sum_{i\in\mathcal{Y},a\in\mathcal{A}} \frac{1}{\gamma_i\sqrt{n_{i,a}}}$ and all $\delta \in [-\gamma_1, \gamma_0]$, we obtain

$$\gamma_0/\gamma_1 = \tilde{n}_1^{1/4}/\tilde{n}_0^{1/4},$$

where the adjusted sample size $\tilde{n}_i = \frac{n_i \Pi_{a\in\mathcal{A}} n_{i,a}}{(\sqrt{\Pi_{a\in\mathcal{A}} n_{i,a}} + 2\alpha \sum_{a\in\mathcal{A}} \sqrt{n_i n_{i,a}})^2}$ for $i \in \{0, 1\}$. From Theorem 3.1, we see how sample sizes of each subgroups are taken into account and how they affect the optimal ratio between class margins. Based on this theorem, we will propose our theoretical framework in Section 4. A closely related derivation has been used in Cao et al. (2019), but their focus is only on the classification error and its generalization. As we will show in Example 3.1, when fairness constraints are also considered, their methods could sometimes perform poorly with respect to the generalization of those constraints. We remark here that if we do not consider the fairness constraints violation, then $\alpha = 0$, and the effective sample sizes degenerate to $\tilde{n}_i = n_i$.

For illustration, we demonstrate the advantage of applying our approach to select margins over directly using the margin selection in Cao et al. (2019) by considering Gaussian models, which is

widely used in machine learning theory (Schmidt et al., 2018; Zhang et al., 2021; Deng et al., 2021). Specifically, our training data follow distribution: $X|Y = 0 \sim \sum_{i=1}^2 \pi_{0,a_i} \mathcal{N}_p(\mu_i, I)$, $X|Y = 1 \sim \sum_{i=1}^2 \pi_{1,a_i} \mathcal{N}_p(\mu_i + \beta^*, I)$. Here, in class $j$, subgroup $a_i$ is drawn with probability $\pi_{j,a_i}$, then, given the sample is from subgroup $a_i$ in class $j$, the data is distributed as a Gaussian random vector. Recall the corresponding training dataset indices of subgroup $a_i$ in class $j$ is denoted as $S_{j,a_i}$, and $|S_{j,a_i}| = n_{j,a_i}$. Consider the case $\alpha = 1$, $\pi_{0,a_1} = \pi_{0,a_2}$, and the following class of classifiers: $\mathcal{F} = \left\{ \mathbf{1}\{\beta^{*\top} x > c\} : c \in \mathbb{R} \right\}$, which is a linear classifier class that contains classifiers differ from each other by a translation in a particular direction.

**Example 3.1** *Given function $f$ and set $S$, let $\mathrm{dist}(f, S) = \min_{x,s \in S} \|f(x) - s\|_2$. Consider two classifiers $\tilde{f}, f \in \mathcal{F}$ such that $\mathrm{dist}(\tilde{f}, S_0)/\mathrm{dist}(\tilde{f}, S_1) = \tilde{n}_1^{1/4}/\tilde{n}_0^{1/4}$ and $\mathrm{dist}(f', S_0)/\mathrm{dist}(f', S_1) = n_0^{-1/4}/n_1^{-1/4}$. Suppose $\|\beta^*\| \gg \sqrt{p \log n}$, $\|\mu_i\| < C$, $(\mu_1^* - \mu_2^*)^\top \beta = 0$, and $\pi_{1,a_2} \le c_1 \pi_{1,a_1}$ for a sufficiently small $c_1 > 0$, then when $n_0, n_1$ are sufficiently large, with high probability we have $\mathcal{M}[\tilde{f}] < \mathcal{M}[f]$. Here $\mathcal{L}_{bal}[f] = \frac{1}{2}\mathbb{P}[f(X)_1 < f(X)_0|Y = 1] + \frac{1}{2}\mathbb{P}[f(X)_0 < f(X)_1|Y = 0]$ denotes the balanced mis-classification error.*

**Remark.** We provide analyses for the 0-1 loss as our ultimate goal is to strike a balance between good **test accuracy** and small **fairness constraints violation**. If we use surrogates such as the softmax-cross-entropy loss for the 0-1 loss in training, our theoretical analyses still stand since we always adjust margins based on the 0-1 loss as our interests are in quantities such as test accuracy. We provide analyses and experiments for DP in the Appendix.

## 4 FLEXIBLE COMBINATION WITH LOGITS-BASED LOSSES

For ease of exposition, we focus solely on the EO constraint hereafter and discuss other constraints in the Appendix. Inspired by the margin trade-off characterized in Section 3, we propose our FIFA approach for **F**lexible **I**mbalance-**F**airness-**A**ware classification that can be easily combined with different types of logits-based losses, and further incorporated into any existing fairness algorithms such as those discussed in Section 5. Recall $\gamma_{i,a}$ is the margin for demographic subgroups in Theorem 3.1, and it could be written as $\gamma_{i,a} = \gamma_i + \delta_{i,a}$ and $\delta_{i,a} \ge 0$ (since $\gamma_i = \min\{\gamma_{i,a_1}, \gamma_{i,a_2}\}$, also see Fig. 3 for illustration), hence the middle term of Eq. (3) can be further upper bounded by the last term in Eq. (2). The final upper bound in Eq. (2) is indeed sufficient for obtaining the margin trade-off between classes. Nonetheless, if we want to further enforce margins for each demographic subgroup in each class, we need to use the refined bound.

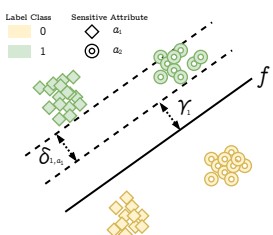

**Figure 3:** Illustration of $\delta_{i,a}$ and the margin $\gamma$ of classifier: $\delta_{1,a_1}$ is set to be non-negative and $\delta_{1,a_2}$ is set to be zero as the subgroup $(1, a_2)$ is closer to the decision boundary than $(1, a_1)$.

Specifically, in Section 3, we have identified a way to select $\gamma_0/\gamma_1$, based on which we propose to enforce margins for each demographic subgroup's training set $S_{i,a}$ of the form

$$\gamma_{i,a} = C/\tilde{n}_i^{1/4} + \delta_{i,a}, \tag{3}$$

where $\delta_{i,a}$ and $C$ are all non-negative tuning parameters. In light of the trade-off between the class margins $\gamma_0/\gamma_1 = \tilde{n}_1^{1/4}/\tilde{n}_0^{1/4}$, we can set $\gamma_i$ of the form $C/\tilde{n}_i^{-1/4}$. Given $\gamma_{i,a} \ge \gamma_i$, a natural choice for margins for subgroups is Eq. (3).

**How to select $\delta_{i,a}$?** Knowing the form of margins from the preceding discussions, an outstanding question remains: how to select $\delta_{i,a}$ for imbalanced datasets? Let $\mathcal{Y} = \{0, 1\}$ and $\mathcal{A} = \{a_1, a_2\}$, within each class $i$, we identify $S_{i,a}$ with the largest cardinality $|S_{i,a}|$ and set the corresponding $\delta_{i,a} = 0$. The remaining $\delta_{i,\mathcal{A}\setminus a}$ are tuned as a non-negative parameter. As a further illustration, without loss of generality, assume for all $i$, $|S_{i,a_1}| \ge |S_{i,a_2}|$. Thus selected $\{\delta_{i,a}\}_{i,a}$ ensures the upper bound in the middle of Eq. (2) is tighter in the sense that for any $\delta > 0$,

$$\sum_{i \in \mathcal{Y}} \left( \frac{1}{\gamma_i \sqrt{n_{i,a_1}}} + \frac{1}{(\gamma_i + \delta)\sqrt{n_{i,a_2}}} \right) \le \sum_{i \in \mathcal{Y}} \left( \frac{1}{(\gamma_i + \delta)\sqrt{n_{i,a_1}}} + \frac{1}{\gamma_i \sqrt{n_{i,a_2}}} \right).$$

In the Appendix, we will present how to choose $\delta_{i,a}$'s when there are multiple demographic groups. Briefly speaking, our results similar to the above inequality are proved by an application of the

rearrangement inequality. Simple as it is, the high-level view is meaningful – the decision boundaries of a fair predictor should be farther away from the less-frequent subgroup than the more-frequent subgroup to ensure better fairness generalization.

**Flexible imbalance-fairness-aware (FIFA) approach.** We will demonstrate how to apply the above motivations to design better margin losses. Loosely speaking, we consider a logits-based loss

$$\ell((x, y); f) = \ell(f(x)_y, \{f(x)_i\}_{i \in \mathcal{Y} \backslash y}),$$

which is non-increasing with respect to its first coordinate if we fix the second coordinate. Such losses include (i). 0-1 loss: $\mathbf{1}\{f(x)_y < \max_{i \in \mathcal{Y} \backslash y} f(x)_i\}$. (ii). Hinge loss: $\max\{\max_{i \in \mathcal{Y} \backslash y} f(x)_i - f(x)_y, 0\}$. (iii). Softmax-cross-entropy loss: $-\log e^{f(x)_y}/(e^{f(x)_y} + \sum_{i \neq y} e^{f(x)_i})$.

Our flexible imbalance-fairness-aware (FIFA) approach modifies the above losses by enforcing margin of the form in Eq. (3). Specifically, we use the following loss function ***during training***

$$\ell_{\text{FIFA}}((x, y, a); f) = \ell(f(x)_y - \Delta_{y,a}, \{f(x)_i\}_{i \in \mathcal{Y} \backslash y}), \tag{4}$$

where $\Delta_{i,a} = C/\tilde{n}_i^{1/4} + \delta_{i,a}$. We remark here $\ell_{\text{FIFA}}((x, y, a); f)$ is used only during training phase, where we allow access to sensitive attribute $a$. In the test time, we only need to use $f$ but ***not*** $a$.

## 5 EXAMPLE: COMBINING FIFA WITH REDUCTIONS-BASED FAIR ALGORITHMS

---

**Algorithm 1** FIFA Combined Grid Search

---

**Input:** Training data set $\{x_i, y_i, a_i\}_{i=1}^n$, fairness tolerance $\epsilon$, margins $\{\Delta_{y,a}\}_{y,a}$, a classifier $h(\cdot; \theta)$.
**Output:** the learnt classifier $h^*$.
 1: Load training data to **GridS**.
 2: **GridS** produces a set of reduction-labels $\hat{y}_{\text{train}}$ and a set of sample weights $w_{\text{train}}$ based on the type of fairness constraint and tolerance $\epsilon$.
 3: **for** $t = 1, 2, \ldots, T$ **do**
 4:     Compute the FIFA loss $\ell_{\text{FIFA}}$ via (4) using reduction-labels $\hat{y}_{\text{train}}$ (in mini-batches).
 5:     Update $\theta$ in $h$ using back-propagation.
 6:     Logging training metrics using true labels $\{y_i\}_{i=1}^n$ and attributes $\{a_i\}_{i=1}^n$.
 7: **end for**

---

In this section, we demonstrate the power of our approach by combining it with a popular reduction-based fair classification algorithm (Agarwal et al., 2018) *as an example*. In Section 6, we show that incorporating our approach can bring a significant gain in terms of both combined loss and fairness generalization comparing with directly applying their method in vanilla models trained with softmax-cross-entropy losses. The reduction approach proposed in Agarwal et al. (2018) has two versions: (i). **Exponentiated gradient** (**ExpGrad**) that produces a randomized classifier; and (ii). **Grid search** (**GridS**) that produces a deterministic classifier. Our approach can be combined with both. From a high level point, the above two methods are mainly based on putting the fairness constraint as a penalty along with the objective loss function then perform a min-max optimization. The only difference is that the first algorithm **ExpGrad** aims to produce a randomized classifier and **GridS** aims to produce a deterministic classifier. To incorporate our framework with the two algorithms above, we only need to slightly modify the error function by adding a margin related term.

**Exponentiated gradient (ExpGrad).** We first briefly describe the algorithm here. For $\mathcal{Y} = \{0, 1\}$, by Agarwal et al. (2018), EO constraints could be rewritten as $M\mu(h) \leq c$ for certain $M$ and $c$, where $\mu_j(h) = \mathbb{E}[h(X)|E_j]$ for $j \in \mathcal{J}$, $M \in \mathbb{R}^{|\mathcal{K}| \times |\mathcal{J}|}$, and $c \in \mathbb{R}^{\mathcal{K}}$. Here, $\mathcal{K} = \mathcal{A} \times \mathcal{Y} \times \{+, -\}$ ($+, -$ impose positive/negative sign so as to recover $|\cdot|$ in constraints) and $\mathcal{J} = (\mathcal{A} \cup \{*\}) \times \{0, 1\}$. $E_{(a,y)} = \{A = a, Y = y\}$ and $E_{(*,y)} = \{Y = y\}$. Let $\text{err}(h) = \mathbb{P}(h(X) \neq Y)$, instead of considering $\min_{h \in \mathcal{H}} \text{err}(h)$ such that $M\mu(h) \leq c$, **ExpGrad** obtains the best *randomized classifier*, by sampling a classifier $h \in \mathcal{H}$ from a distribution over $\mathcal{H}$. Formally, this optimization can be formulated as $\min_{Q \in \Delta_{\mathcal{H}}} \text{err}(Q)$ such that $M\mu(Q) \leq c$, where $\text{err}(Q) = \sum_{h \in \mathcal{H}} Q(h) \text{err}(h)$, $\mu(Q) = \sum_{h \in \mathcal{H}} Q(h)\mu(h)$, $Q$ is a distribution over $\mathcal{H}$, and $\Delta_{\mathcal{H}}$ is the collection of distributions on $\mathcal{H}$. Let us further use $\widehat{\text{err}}(Q)$ and $\hat{\mu}(Q)$ to denote the empirical versions and also allows relaxation on constraints by using $\hat{c} = c + \epsilon$, where $\hat{c}_k = c_k + \epsilon_k$ for relaxation $\varepsilon_k \geq 0$. By classic optimization theory, it could be transferred to a saddle point problem, and Agarwal et al. (2018) aims to solve the following prime dual problems simultaneously for $L(Q, \lambda) = \widehat{\text{err}}(Q) + \lambda^\top(M\hat{\mu}(Q) - \hat{c})$:

$$(\mathbf{P}) : \min_{Q \in \Delta} \max_{\lambda \in \mathbb{R}_+^{|\mathcal{K}|}, \|\lambda\|_1 \leq B} L(Q, \lambda), \qquad (\mathbf{D}) : \max_{\lambda \in \mathbb{R}_+^{|\mathcal{K}|}, \|\lambda\|_1 \leq B} \min_{Q \in \Delta} L(Q, \lambda).$$

To summarize, **ExpGrad** takes training data $\{(x_i, y_i, a_i)\}_{i=1}^n$, function class $\mathcal{H}$, constraint parameters $M, \hat{c}$, bound $B$, accuracy tolerance $v > 0$, and learning rate $\eta$ as inputs and outputs $(\hat{Q}, \hat{\lambda})$, such

that $L(\hat{Q}, \hat{\lambda}) \leq L(Q, \hat{\lambda}) + \nu$ for all $Q \in \Delta_{\mathcal{H}}$ and $L(\hat{Q}, \hat{\lambda}) \leq L(\hat{Q}, \lambda) - \nu$ for all $\lambda \in \mathbb{R}_+^{|\mathcal{K}|}, \|\lambda\|_1 \leq B$, and $(\hat{Q}, \hat{\lambda})$ is called a $\nu$-approximate saddle point. As implemented in Agarwal et al. (2018), $\mathcal{H}$ roughly consists of $h(x) = \mathbf{1}\{f(x)_1 \geq f(x)_0\}$ for $f \in \mathcal{F}$ (in fact, a smoothed version is considered in Agarwal et al. (2018)) and gives

$$\widehat{\mathrm{err}}(Q) = \sum_{h \in \mathcal{H}} \hat{\mathbb{P}}(h(X) \neq Y)Q(h) = \hat{\mathbb{P}}(f(X)_Y < f(X)_{\{0,1\}\backslash Y})Q(h).$$

To combine our approach, we consider optimizing $\widehat{\mathrm{err}}^{\mathrm{new}}(Q) = \sum_{h \in \mathcal{H}} \hat{\mathbb{P}}(f(X)_Y - \Delta_{Y,A} \leq f(X)_{\{0,1\}\backslash Y})Q(h)$, such that $M\hat{\mu}^{\mathrm{new}}(Q) \leq \hat{c}$, where $\hat{\mu}^{\mathrm{new}}(Q) = \sum_{h \in \mathcal{H}} Q(h)\hat{\mu}^{\mathrm{new}}(f)$ and $\hat{\mu}_j^{\mathrm{new}}(f) = \hat{\mathbb{P}}(f(X)_Y - \Delta_{Y,A} > f(X)_{\{0,1\}\backslash Y}|E_j)$. We can modify **ExpGrad** to optimize prime dual problems simultaneously for $L^{\mathrm{new}}(Q, \lambda) = \widehat{\mathrm{err}}^{\mathrm{new}}(Q) + \lambda^\top(M\hat{\mu}^{\mathrm{new}}(Q) - \hat{c})$. In practice, while Section 3 is motivated for deterministic classifiers, FIFA works for the randomized version too – the modified **ExpGrad** can be viewed as encouraging a distribution $Q$ that puts more weights on classifiers with a certain type of margin trade-off between classes. Moreover, the modified algorithm enjoys similar convergence guarantee as the original one.

**Theorem 5.1** *Let $\rho = \max_f \|M\hat{\mu}^{\mathrm{new}}(f) - \hat{c}\|_\infty$. For $\eta = \nu/(2\rho^2 B)$, the modified **ExpGrad** will return a $\nu$-approximate saddle point of $L^{\mathrm{new}}$ in at most $4\rho^2 B^2 \log(|\mathcal{K}| + 1)/\nu^2$ iterations.*

**Grid search (GridS).** When the number of constraints is small, e.g., when there are only few sensitive attributes, one may directly perform a grid search on the $\lambda$ vectors to identify the *deterministic classifier* that attains the best trade-off between accuracy and fairness. In practice, **GridS** is preferred for larger models due to its memory efficiency, since **ExpGrad** needs to store all intermediate models to compute the randomized classifier at prediction time, which is less feasible for over-parameterized models. We describe our flexible approach in Algorithm 1 that combines with **GridS** used in practice in the official code base `FairLearn` (Bird et al., 2020).

## 6 EXPERIMENTS

We now use our flexible approach on several datasets in the classification task with a sensitive attribute. Although our method is proposed for over-parameterized models, it can also boost the performance on small models. Depending on the specific dataset and model architectures, we use either the grid search or the exponentiated gradient method developed by Agarwal et al. (2018) as fairness algorithms to enforce the fairness constraints, while adding our FIFA loss in the inner training loop. Note that our method can be combined with other fairness algorithms.

**Datasets.** We choose both a large image dataset and two simpler datasets. We use the official train-test split of these datasets. More details and statistics are in the Appendix. **(i). CelebA** ((Liu et al., 2015)): the task is to predict whether the person in the image has blond hair or not where the sensitive attribute is the gender of the person. **(ii). AdultIncome** ((Dua & Graff, 2017)): the task is to predict whether the income is above 50K per year, where the sensitive attribute is the gender. We also use the new AdultIncome (from California in 2021) introduced by Ding et al. (2021), where the sensitive attribute is the race. **(iii). DutchConsensus** ((voor de Statistiek , Statistics Netherlands)): the task is predict whether an individual has a prestigious occupation and the sensitive attribute is the gender. Both AdultIncome and DutchConsensus datasets are also used in Agarwal et al. (2018).

**Method.** Due to computational feasibility (**ExpGrad** needs to store all intermediate models at prediction time), we combine Grid Search with FIFA for the CelebA dataset and ResNet-18 and use both Grid Search and Exponentiated Gradient on the AdultIncome with logistic regression. Besides $C$ and $\delta_{i,a}$, we also treat $\alpha$ as tuning parameters (in Eq. (2)). We then perform hyper-parameter sweeps on the grids (if used) over $C$, $\delta_{i,a}$ and $\alpha$ for FIFA, and grids (if used) for vanilla training (combine fairness algorithms with the vanilla softmax-cross-entropy loss). More details are included in the Appendix.

**Evaluation and Generalization.** When evaluating the model, we are mostly interested in the generalization performance measured by a *combined loss* that take into consideration both fairness violation and balanced error. We define the combined loss as $\mathcal{L}_{\mathrm{cb}}[f] = \frac{1}{2}\mathcal{L}_{\mathrm{bal}}[f] + \frac{1}{2}\mathcal{L}_{\mathrm{fv}}[f]$, which favors those classifiers that have a equally well-performance in terms of classification and fairness. We consider both the value of the combined loss evaluated on the test set $\mathcal{S}_{\mathrm{test}}$, and the *generalization error* for a loss $\mathcal{L}$ is defined as $\mathrm{GenErr}[\mathcal{L}, f] = |\mathcal{L}[f](\mathcal{S}_{\mathrm{test}}) - \mathcal{L}[f](\mathcal{S}_{\mathrm{train}})|$.

| Fairness Tolerance $\epsilon$ | | **0.01** | | | **0.05** | | | **0.10** | | |
| Method | | FIFA | LDAM | Vanilla | FIFA | LDAM | Vanilla | FIFA | LDAM | Vanilla |
|---|---|---|---|---|---|---|---|---|---|---|
| **Combined Loss** | Test | **6.71%** | 7.29% | 14.01% | **6.34%** | 7.38% | 13.05% | **6.54%** | 7.34% | 16.71% |
| | Gen Error | **0.66%** | 2.07% | 6.87% | **0.88%** | 1.91% | 4.21% | **0.62%** | 1.14% | 7.82% |
| **Fairness Violation** | Test | **2.75%** | 5.39% | 20.29% | **3.29%** | 5.57% | 17.92% | **2.65%** | 2.96% | 26.15% |
| | Gen Error | **2.57%** | 3.07% | 13.59% | **0.66%** | 3.00% | 8.47% | **0.46%** | 3.97% | 14.78% |
| **Balanced Error** | Test | 10.67% | 9.20% | **7.74%** | 9.39% | 9.19% | **8.17%** | 10.43% | 11.72% | **7.27%** |
| | Gen Error | 1.25% | 1.07% | **0.15%** | 1.10% | 0.83% | **0.05%** | 1.70% | 1.68% | **0.85%** |

**Table 1:** Grid search with EO constraint on CelebA dataset (Liu et al., 2015) using ResNet-18, best results with respect to test combined loss among sweeps of hyper-parameters are shown. As an interesting special case of our FIFA method, we note that although the LDAM method improves the performance compared with vanilla GS, it is not as effective as our method.

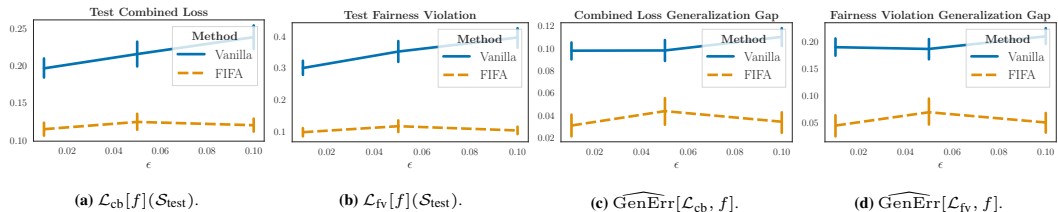

(a) $\mathcal{L}_{\text{cb}}[f](\mathcal{S}_{\text{test}})$.     (b) $\mathcal{L}_{\text{fv}}[f](\mathcal{S}_{\text{test}})$.     (c) $\widehat{\text{GenErr}}[\mathcal{L}_{\text{cb}}, f]$.     (d) $\widehat{\text{GenErr}}[\mathcal{L}_{\text{fv}}, f]$.

**Figure 4:** Loss on test set (4a,4b) and generalization gaps (4c,4d) of the combined loss and the fairness loss on CelebA dataset. We repeat the experiment for 20 times using the hyper-parameters corresponding to the best-performing models in Table 1. Solid blue line marks the grid search combined with vanilla training whereas dashed orange line marks grid search combined with the FIFA loss. We also plot 95% confidence band based on 20 repeated runs. We observe that our method FIFA has significantly better generalization performance in terms of both smaller losses on the test set as well as narrower generalization gaps.

## 6.1 EFFECTIVENESS OF FIFA ON OVER-PARAMETERIZED MODELS

In this subsection, we thoroughly analyze the results from applying FIFA to over-parameterized models for the CelebA dataset using ResNet-18. We use the grid search algorithm with fairness violation tolerance parameter $\epsilon \in \{0.01, 0.05, 0.1\}$ (with a little abuse of notation) for all constraints. We perform sweeps on hyper-parameters $C \in [0, 0.01]$, $\alpha \in [0, 0.01]$, $\delta_{0,\text{Male}}, \delta_{1,\text{Male}} \in [0, 0.01]$, and $\delta_{0,\text{Female}} = \delta_{1,\text{Female}} = 0$. As a special case that may be of interest, when $\alpha = 0$ and $\delta_{0,\text{Male}} = \delta_{1,\text{Male}} = 0$, the FIFA loss coincides with the LDAM loss proposed in Dua & Graff (2017), with one common hyper-parameter $C \in [0, 0.01]$. We log the losses on the whole training and test set. We summarize our main findings below and give more details in the Appendix including experiments with DP constraints, delayed-reweighting (DRW, (Cao et al., 2019)), and reweighting methods.

**Logits-based methods improve fairness generalization.** We summarize the best results for each method under different tolerance parameter $\epsilon$ in Table 1. Note that the actual violation may exceed the tolerance $\epsilon$ on test data. We note that both FIFA and LDAM *significantly* improve the test performance of both combined loss and fairness violation among all three choices of $\epsilon$, while having comparable training performance (omitted in the table). Interestingly, directly applying reductions-based method using a vanilla model in the inner loop (the "Vanilla" columns) seems inferior, likely due to the imbalance across subgroups. This implies the effectiveness and necessity of using logits-based methods to ensure a better fairness generalization.

**FIFA accommodates for both fairness generalization and dataset imbalance.** Although both logits-based method improve generalization as seen in Table 1, our method FIFA has significantly better generalization performance compared with LDAM, especially in terms of fairness violation. For example, when $\epsilon = 0.01$ and $0.05$, FIFA achieves a test fairness violation that is at least 2% smaller compared with LDAM. This further demonstrates the importance of our theoretical motivations.

**Improvements of generalization are two-fold for FIFA.** When it comes to generalization, two relevant notions are often used, namely the absolute performance on the test set, and also the generalization error between the training and test set. We compute the generalization error in Table 1 for both combined loss and fairness violation. We observe that FIFA generally dominates LDAM and vanilla in terms of both test performance and generalization error. We further illustrate this behavior in Fig. 4, where we give 95% confidence band over randomness in training. We note that our FIFA significantly outperforms vanilla in a large margin in terms of both generalization notions, and the improvements are mostly due to better fairness generalization. In fact, as suggested by the similarity in the shapes of curves between Fig. 4c and Fig. 4d, fairness generalization dominates classification generalization, and thus improvements in fairness generalization elicit more prominently overall.

| Dataset | | AdultIncome | | | | DutchConsensus | | | |
|---|---|---|---|---|---|---|---|---|---|
| Metric | | Combined Loss | | Fairness Violation | | Combined Loss | | Fairness Violation | |
| $\epsilon$ | Method | Train | Test | Train | Test | Train | Test | Train | Test |
| 0.01 | FIFA | 15.7217% | **13.4812%** | 7.8863% | **2.7776%** | 12.8013% | **13.1686%** | 3.7220% | **4.3532%** |
| | Vanilla | 13.9561% | 14.3001% | 6.7861% | 6.7475% | 12.8444% | 13.2267% | 3.7935% | 4.4323% |
| 0.05 | FIFA | 13.5634% | **13.5491%** | 5.7088% | **4.9315%** | 12.8820% | **13.2228%** | 3.8525% | **4.4323%** |
| | Vanilla | 14.4697% | 14.8647% | 7.5962% | 7.8575% | 12.8818% | 13.2236% | 3.8525% | 4.4323% |
| 0.10 | FIFA | 13.5857% | **13.9043%** | 6.1217% | **5.9689%** | 12.8717% | **13.1748%** | 3.8326% | **4.3532%** |
| | Vanilla | 15.5342% | 15.9387% | 9.7514% | 10.0750% | 12.8757% | 13.2099% | 3.8392% | 4.4059% |

**Table 2:** Exponentiated gradient with EO constraint on the AdultIncome and DutchConsensus datasets using logistc regression (as a one-layer neural net), best results with respect to test combined loss among sweeps of hyper-parameters are shown.

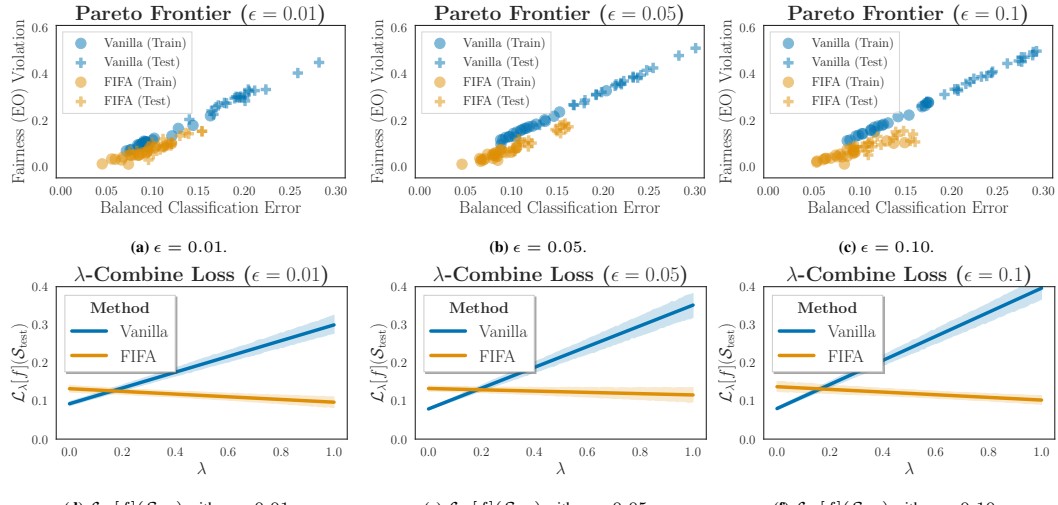

**(a)** $\epsilon = 0.01$.     **(b)** $\epsilon = 0.05$.     **(c)** $\epsilon = 0.10$.

**(d)** $\mathcal{L}_\lambda[f](\mathcal{S}_{\text{test}})$ with $\epsilon = 0.01$.     **(e)** $\mathcal{L}_\lambda[f](\mathcal{S}_{\text{test}})$ with $\epsilon = 0.05$.     **(f)** $\mathcal{L}_\lambda[f](\mathcal{S}_{\text{test}})$ with $\epsilon = 0.10$.

**Figure 5:** Pareto frontiers of the balanced loss ($\mathcal{L}_{\text{bal}}$) and fairness loss ($\mathcal{L}_{\text{fv}}$) of CelebA using ResNet-18 with grid search combined with FIFA and vanilla softmax-cross-entropy loss respectively. (a)-(c) exhibit Porento frontier and (d)-(f) illustrate $\lambda$-combined loss. Best-performing hyper-parameters from Table 1 are used, where each configuration is repeated 20 times independently. Here blue and orange markers correspond to vanilla and FIFA respectively, and circular and cross markers correspond to training and testing metrics respectively. We observe that our FIFA method is effective in significantly lowering the Pareto frontier comparing with the vanilla method, implying that FIFA mitigates fairness generalization issues as seen in Figure 1.

**Towards a more efficient Pareto frontier.** In Fig. 5(a)-(c) we plot the Pareto frontier of balanced classification error ($\mathcal{L}_{\text{bal}}$) and fairness violation ($\mathcal{L}_{\text{fv}}$) for all three choices of $\epsilon$. In practice, one may be interested in a specific convex combination of the fairness violation and balanced error. We thus consider $\lambda$-weighted combined loss $\mathcal{L}_\lambda = \lambda \mathcal{L}_{\text{fv}} + (1 - \lambda) \mathcal{L}_{\text{bal}}$ with $\lambda \in [0, 1]$ being a user-specific weight. In Fig. 5(d)-(f), we compute $\mathcal{L}_\lambda$ for a grid of 100 values of $\lambda$ under the same setup. We observe that FIFA with GridS achieves frontiers that are lower and more centered compared with those trained in vanilla losses with GridS. Furthermore, for most of the combining weight $\lambda$, FIFA achieves better test performance.

## 6.2 EFFECTIVENESS OF FIFA ON SMALLER DATASETS AND MODELS

We use logistic regression (implemented as a one-layer neural net) for the AdultIncome and Dutch-Consensus datasets with similar sweeping procedure are similar to those described in Section 6.1.

**Results.** We tabulate the best-performing models (in terms of test combined loss) among sweeps in Table 2 and include more details in the Appendix. The observations are similar as in Section 6.1: FIFA outperforms vanilla on both dataset across three different tolerance parameter $\epsilon$; since the datasets are much simpler in this case, the improvements are less significant.

## 7 DISCUSSIONS AND CONCLUSIONS

Generalization (especially in over-parameterized models) has always been an important and difficult problem in machine learning research. In this paper, we set out the first exposition in the study of the generalization of *fairness constraints* that has previously been overlooked. Our theoretically-backed FIFA approach is shown to mitigate poor fairness generalization observed in vanilla models large or small. We leave a more fine-grained analysis of the margins to the future work.

## ACKNOWLEDGEMENTS

The research of Jiayao Zhang is partially supported by DARPA FA8750-19-2-1004, IARPA 2019-19051600006, and ONR N00014-19-1-2620. The research of Linjun Zhang is partially supported by NSF DMS-2015378. The research of James Zou is partially supported by funding from NSF CAREER and the Sloan Fellowship. We would also like to thank the support of Kaiser Permanente Washington Health Research Institute and funding R01 MH125821.

## ETHICS STATEMENT

We mainly consider specific type of fairness constraints such as equalized odds, equalized opportunity, and demographic parity. That may be a restriction in ethics. We hope our argument could be generalized to other fairness notions in the future.

## REPRODUCIBILITY STATEMENT

Our code is available to the public on GitHub at `https://github.com/zjiayao/fifa-iclr23`.

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

# Appendix

## A  OMITTED DERIVATION

In this section, we will talk about several missing details in the main context.

### A.1  EXTENSION TO MULTI-CLASSES AND MULTI-GROUPS FOR EQUALIZED ODDS

We discussed the case that $\mathcal{Y} = \{0, 1\}$ and $\mathcal{A} = \{a_1, a_2\}$ in the main context. In this subsection, we discuss the extension to multi-classes and multi-groups.

First, we extend the case to $\mathcal{Y} = \{0, 1\}$ and $|\mathcal{A}| \geq 2$. In that case, we can consider the constraints

$$\sum_{a,a' \in \mathcal{A}, a \neq a'} \sum_{i \in \mathcal{Y}} |\mathbb{P}(h(X) = i | Y = i, A = a) - \mathbb{P}(h(X) = i | Y = i, A = a')|,$$

regardless of the order of $(a, a')$, and there are $\binom{|\mathcal{A}|}{2}$ pairs of $(a, a')$'s.

Thus, our performance criteria $\mathcal{M}_{\text{multi-groups}}[f]$ can be taken as:

$$\mathcal{L}_{\text{bal}}[f] + \alpha \sum_{a,a' \in \mathcal{A}, a \neq a'} \sum_{i \in \mathcal{Y}} |\mathbb{P}(h(X) = i | Y = i, A = a) - \mathbb{P}(h(X) = i | Y = i, A = a')|.$$

Then, via Lemma A.2, for all $f \in \mathcal{F}$

$$\mathcal{M}_{\text{multi-groups}}[f] \lesssim \sum_{i \in \mathcal{Y}} \frac{1}{\gamma_i} \sqrt{\frac{C(\mathcal{F})}{n_i}} + \sum_{i \in \mathcal{Y}, a \in \mathcal{A}} \frac{2\alpha(|A| - 1)}{\gamma_{i,a}} \sqrt{\frac{C(\mathcal{F})}{n_{i,a}}}$$

$$\leq \sum_{i \in \mathcal{Y}} \frac{1}{\gamma_i} \sqrt{\frac{C(\mathcal{F})}{n_i}} + \sum_{i \in \mathcal{Y}, a \in \mathcal{A}} \frac{2\alpha(|\mathcal{A}| - 1)}{\gamma_i} \sqrt{\frac{C(\mathcal{F})}{n_{i,a}}}.$$

We overload the notation $\bar{\alpha} = 2\alpha(|\mathcal{A}| - 1)$, then we have

$$\mathcal{M}_{\text{multi-groups}}[f] \lesssim \sum_{i \in \mathcal{Y}} \frac{1}{\gamma_i} \sqrt{\frac{C(\mathcal{F})}{n_i}} + \sum_{i \in \mathcal{Y}, a \in \mathcal{A}} \frac{\bar{\alpha}}{\gamma_i} \sqrt{\frac{C(\mathcal{F})}{n_{i,a}}}. \tag{A.1}$$

Notice the difference between Eq. (A.1) and Eq. (2) is that in Eq. (A.1), $|\mathcal{A}| \geq 2$. Thus, by similar proof as in Theorem A.1, we can obtain

$$\gamma_0/\gamma_1 = \tilde{n}_0^{-1/4}/\tilde{n}_1^{-1/4},$$

where the adjusted sample size

$$\tilde{n}_i = \frac{n_i \Pi_a n_{i,a}}{(\sqrt{\Pi_a n_{i,a}} + \bar{\alpha} \sum_{j \in \mathcal{A}} \sqrt{n_i \Pi_{a \in \mathcal{A} \setminus j} n_{i,a}})^2},$$

for $i \in \{0, 1\}$.

Given the results above, for multiple classes with multiple groups, for $i, j \in \mathcal{Y}$,

$$\gamma_i/\gamma_j = \tilde{n}_i^{-1/4}/\tilde{n}_j^{-1/4},$$

where the adjusted sample size $\tilde{n}_i = \frac{n_i \Pi_a n_{i,a}}{(\sqrt{\Pi_a n_{i,a}} + \bar{\alpha} \sum_{j \in \mathcal{A}} \sqrt{n_i \Pi_{a \in \mathcal{A} \setminus j} n_{i,a}})^2}$ for $i \in \mathcal{Y}$.

**FIFA for multi-classes and multi-groups.**  We will demonstrate how to apply the above motivations to design better margin losses. Consider a logits-based loss

$$\ell((x, y); f) = \ell(f(x)_y, \{f(x)_i\}_{i \in \mathcal{Y} \setminus y}),$$

which is non-increasing with respect to its first coordinate if we fix the second coordinate.

Our flexible imbalance-fairness-aware (FIFA) approach modifies the above losses during training

$$\ell_{\text{FIFA}}((x, y, a); f) = \ell(f(x)_y - \Delta_{y,a}, \{f(x)_i\}_{i \in \mathcal{Y} \setminus y}) \tag{A.2}$$

where $\Delta_{i,a} = C/\tilde{n}_i^{1/4} + \delta_{i,a}$, and $\delta_{i,a} \geq 0$. The specific assignment of $\delta_{i,a} \geq 0$ is described in Section A.2.

## A.2 Assignment of $\delta_{i,a}$ for multi-groups

In this subsection, we describe how to choose $\delta_{i,a}$ for multiple demographic groups. Recall in Section A.1, for multiple demographic groups,

$$\mathcal{M}_{\text{multi-groups}}[f] \lesssim \sum_{i \in \mathcal{Y}} \frac{1}{\gamma_i} \sqrt{\frac{C(\mathcal{F})}{n_i}} + \sum_{i \in \mathcal{Y}, a \in \mathcal{A}} \frac{\bar{\alpha}}{\gamma_{i,a}} \sqrt{\frac{C(\mathcal{F})}{n_{i,a}}}. \tag{A.3}$$

Given $\gamma_{i,a} \geq \gamma_i$, similar as in the main context, we can take $\gamma_{i,a} = \gamma_i + \delta_{i,a}$, where $\delta_{i,a} \geq 0$ are tuning parameters.

Assume there are $k$ groups. Let us first ordering $|S_{i,a}|$ by in a decreasing order. Without loss of generality, $|S_{i,a_1}| \geq |S_{i,a_2}| \geq \cdots \geq |S_{i,a_k}|$. Thus, when tune the parameters $\delta_{i,a}$'s, we set $\delta_{i,a_1} = 0$ (or we can randomly choose other $a$'s to set $\delta_{i,a} = 0$ if there are ties and $|S_{i,a}| = |S_{i,a_1}|$, but for simplicity, we ignore this case), and we make sure $\delta_{i,a_1} \leq \delta_{i,a_2} \leq \cdots \delta_{i,a_k}$. This is optimal in the sense that if there are $k$ constants $0 = \delta_1 \leq \delta_2 \leq \delta_3 \leq \cdots \leq \delta_k$, then

$$\sum_{i \in \mathcal{Y}, a_j \in \mathcal{A}} \frac{\bar{\alpha}}{\gamma_i + \delta_{i,a_j}} \sqrt{\frac{C(\mathcal{F})}{n_{i,a_j}}} \leq \sum_{i \in \mathcal{Y}, a_j \in \mathcal{A}} \frac{\bar{\alpha}}{\gamma_i + \delta_{i,a_{\sigma(j)}}} \sqrt{\frac{C(\mathcal{F})}{n_{i,a_j}}},$$

where $\sigma(\cdot)$ is a permutation. In other words, our way to assign $\delta_{i,a}$'s can make the upper boupnd on RHS of Eq. (A.3) optimal. This is a direct application of rearrangement inequality, see Lemma A.1.

**Lemma A.1** *For $x_1 \leq x_2 \leq \cdots \leq x_k$, $y_1 \leq y_2 \leq \cdots \leq y_k$, any permutation $\sigma(\cdot)$*

$$x_k y_1 + x_{k-1} y_2 + \cdots + x_1 y_k \leq x_{\sigma(1)} y_1 + x_{\sigma(2)} y_2 + \cdots + x_{\sigma(k)} y_k \leq x_1 y_1 + x_2 y_2 + \cdots + x_k y_k.$$

## A.3 Derivation for other fairness notions

In this subsection, we consider theory-inspired derivation for other fairness notions. For simplicity, we still focus on the case that $\mathcal{Y} = \{0, 1\}$ (this is necessary for EqOpt) and $\mathcal{A} = \{a_1, a_2\}$. Given the derivation in this subsection, we can further derive FIFA for other fairness notions as in Section A.1.

**Equalized opportunity.** Specifically, for EqOpt, the aim is:

$$\min_f \mathcal{L}_{\text{bal}}[f]$$
$$\text{s.t. } \forall y \in \mathcal{Y}, a \in \mathcal{A}, \ \mathbb{P}(h(X) = 1 | Y = 1, A = a) = \mathbb{P}(h(X) = 1 | Y = 1),$$

This is a simple version of EO in some sense. Directly using the derivation in Section A.1, we have

$$\gamma_0 / \gamma_1 = n_0^{-1/4} / \tilde{n}_1^{-1/4},$$

where the adjusted sample size

$$\tilde{n}_1 = \frac{n_1 n_{1,a_1} n_{1,a_2}}{(\sqrt{n_{1,a_1} n_{1,a_2}} + 2\alpha(\sqrt{n_1 n_{1,a_2}} + \sqrt{n_1 n_{1,a_1}}))^2}.$$

**Demographic parity.** Similar to EO, for DP, the optimization aims to:

$$\min_f \mathcal{L}_{\text{bal}}[f]$$
$$s.t. \ \forall y \in \mathcal{Y}, a \in \mathcal{A}, \ \mathbb{P}(h(X) = y | A = a) = \mathbb{P}(h(X) = y).$$

In this setting, we no longer can expect all the training samples are perfectly classified and fairness constraints violation is perfectly satisfied because there exists fairness and accuracy trade-off in training phase for DP. However, in real application, people always adopt relaxation in fairness constraints, i.e. $|\mathbb{P}(h(X) = y | A = a) - \mathbb{P}(h(X) = y)| < \epsilon$ for some $\epsilon > 0$ (if there are only two groups, one can alternatively use $|\mathbb{P}(h(X) = y | A = a_1) - \mathbb{P}(h(X) = y) | A = a_2| \leq \epsilon$). When $\epsilon$ is large enough (or $\hat{\mathbb{P}}(Y = 1 | A = a_1)$ is close to $\hat{\mathbb{P}}(Y = 1 | A = a_2)$), if we use suitable models, similar as in the EO setting, we would expect all the training examples are classified perfectly while

satisfying $|\hat{\mathbb{P}}(h(X) = y|A = a_1) - \hat{\mathbb{P}}(h(X) = y)|A = a_2| \leq \epsilon$. Then, we can use similar techniques to characterize a trade-off between margins.

Specifically, simple calculation leads to $\sum_{i \in \mathcal{Y}} |\mathbb{P}(h(X) = i|A = a_1) - \mathbb{P}(h(X) = i|A = a_2)| \leq \sum_{j \in \{1,2\}} \sum_{i \in \{0,1\}} \mathbb{P}(Y = i|A = a_j)\mathcal{L}_{i,a_j}[f] + I$, where $I$ is a term not related to $f$. Thus, our **_optimization objective_** (not performance criterion) for DP can be taken as:

$$\mathcal{L}_{\text{bal}}[f] + \alpha \sum_{i,a} \mathbb{P}(Y = i|A = a)\mathcal{L}_{i,a}[f], \tag{A.4}$$

for weight $\alpha$. We can use training data to estimate $\mathbb{P}(Y = i|A = a)$. For simplicity, we can also use $\mathcal{L}_{\text{bal}}[f] + \alpha \sum_{i,a} \mathcal{L}_{i,a}[f]$, which shares the same upper bound as in Theorem 3.1 and also implies $\gamma_0/\gamma_1 = \tilde{n}_0^{-1/4}/\tilde{n}_1^{-1/4}$, that will also be used in the experiments in later sections. We should also remark that when $\epsilon$ is too small, our method may not work, this can also be reflected in Table 4, when $\epsilon = 0.01$.

## A.4 OMITTED PROOFS

### A.4.1 PROOF OF THEOREM 3.1

**Theorem A.1 (Restatement of Theorem 3.1)** *With high probability over the randomness of the training data, for $\mathcal{Y} = \{0, 1\}$, $\mathcal{A} = \{a_1, a_2\}$, and for some proper complexity measure of class $\mathcal{F}$, i.e. $C(\mathcal{F})$, for any $f \in \mathcal{F}$,*

$$\mathcal{M}[f] \lesssim \sum_{i \in \mathcal{Y}} \frac{1}{\gamma_i} \sqrt{\frac{C(\mathcal{F})}{n_i}} + \sum_{i \in \mathcal{Y}, a \in \mathcal{A}} \frac{2\alpha}{\gamma_{i,a}} \sqrt{\frac{C(\mathcal{F})}{n_{i,a}}} \leq \sum_{i \in \mathcal{Y}} \frac{1}{\gamma_i} \sqrt{\frac{C(\mathcal{F})}{n_i}} + \sum_{i \in \mathcal{Y}, a \in \mathcal{A}} \frac{2\alpha}{\gamma_i} \sqrt{\frac{C(\mathcal{F})}{n_{i,a}}},$$

*where $\gamma_i$ is the margin of the $i$-th class's sample set $S_i$ and $\gamma_{i,a}$ is the margin of demographic subgroup's sample set $S_{i,a}$.*

This following lemma is the key lemma we will use. Let us define the empirical Rademacher complexity of $\mathcal{F}$ of subgroup/class margin on $S_*$ as

$$\hat{\mathcal{R}}_i(\mathcal{F}) = \frac{1}{n_i} \mathbb{E}_\xi [\sup_{f \in \mathcal{F}} \sum_{j \in S_i} \xi_j [f(x_j)_i - \max_{i' \neq i} f(x_j)_{i'}]],$$

$$\hat{\mathcal{R}}_{i,a}(\mathcal{F}) = \frac{1}{n_{i,a}} \mathbb{E}_\xi [\sup_{f \in \mathcal{F}} \sum_{j \in S_{i,a}} \xi_j [f(x_j)_i - \max_{i' \neq i} f(x_j)_{i'}]],$$

where $\xi_j$ is i.i.d. drawn from a uniform distribution $\{-1, 1\}$.

**Lemma A.2** *Let $\hat{\mathcal{L}}_{\gamma,i}[f] = \mathbb{P}_{X \sim \hat{\mathcal{P}}_i}(\max_{j \neq i} f(X)_j > f(X)_i - \gamma)$ and $\hat{\mathcal{L}}_{\gamma,(i,a)}[f] = \mathbb{P}_{X \sim \hat{\mathcal{P}}_{i,a}}(\max_{j \neq i} f(X)_j > f(X)_i - \gamma)$. With probability at least $1 - \delta$ over the the randomness of the training data, for some proper complexity measure of class $\mathcal{F}$, for any $f \in \mathcal{F}$, $* \in \{i, (i,a)|i \in \mathcal{Y}, a \in \mathcal{A}\}$, and all margins $\gamma > 0$*

$$\mathcal{L}_*[f] \lesssim \hat{\mathcal{L}}_{\gamma,*}[f] + \frac{1}{\gamma} \hat{\mathcal{R}}_*(\mathcal{F}) + \epsilon_*(n_*, \delta, \gamma_*), \tag{A.5}$$

*where $\hat{\mathcal{R}}_*(\mathcal{F})$ is the empirical Rademacher complexity of $\mathcal{F}$ of subgroup/class margin on training dataset corresponding to index set $S_*$, which can be further upper bnounded by $\sqrt{\frac{C(\mathcal{F})}{n_*}}$. Also, $\epsilon_*(n_*, \delta, \gamma_*)$ is usually a low-order term in $n_*$*

*Proof:* This is a direct application of the standard margin-based generalization bound in Kakade et al. (2008). □

**Proof of Theorem A.1.** Notice that all the training samples are classified perfectly by $h$, not only $\mathbb{P}_{(X,Y)\sim\hat{\mathcal{P}}_{\text{bal}}}(h(X) \neq Y) = 0$ is satisfied, we also have that $\mathbb{P}_{(X,Y)\sim\hat{\mathcal{P}}_{i,a_j}}(h(X) \neq Y) = 0$ for all $i \in \mathcal{Y}$ and $a_j \in \mathcal{A}$. We remark here that $\mathbb{P}(h(X) = i|Y = i, A = a) = 1 - \mathbb{P}_{(X,Y)\sim\mathcal{P}_{i,a}}(h(X) \neq Y) = 1 - \mathbb{P}(f(X)_Y < \max_{j\neq Y} f(X)_j)$. Thus, we have

$$\mathcal{L}_{\text{bal}}[f] + \alpha \sum_{i\in\mathcal{Y}} |\mathbb{P}(h(X) = i|Y = i, A = a_1) - \mathbb{P}(h(X) = i|Y = i, A = a_2)|$$

$$\leq \mathcal{L}_{\text{bal}}[f] + \alpha(\mathbb{P}_{(X,Y)\sim\mathcal{P}_{i,a_1}}(h(X) \neq Y) + \mathbb{P}_{(X,Y)\sim\mathcal{P}_{i,a_2}}(h(X) \neq Y)).$$

Notice $\mathcal{L}_{bal} = 1/2 \sum_{i\in\mathcal{Y}} \mathbb{P}_{(X,Y)\sim\mathcal{P}_i}(h(X) \neq Y)$, then, plug in Lemma A.2 and realizing $\hat{\mathcal{L}}_{\gamma,*} = 0$ (all training errors are 0) by our assumption and ignoring low order terms $\epsilon_*(n_*, \delta, \gamma_*)$, we have

$$\mathcal{L}_{\text{bal}}[f] + \alpha(\mathbb{P}_{(X,Y)\sim\mathcal{P}_{i,a_1}}(h(X) \neq Y) + \mathbb{P}_{(X,Y)\sim\mathcal{P}_{i,a_2}}(h(X) \neq Y))$$

$$\leq \sum_{i\in\mathcal{Y}} \frac{1}{2}\mathcal{L}_i[f] + \alpha \sum_{i\in\mathcal{Y},a\in\mathcal{A}} \mathcal{L}_{i,a}[f] \quad (\text{Notice } \mathcal{L}_{i,a}[f] = \mathbb{P}_{(X,Y)\sim\mathcal{P}_{i,a}}(h(X) \neq Y))$$

$$\lesssim \sqrt{C(\mathcal{F})}(\sum_{i\in\mathcal{Y}} \frac{1}{\gamma_i\sqrt{n_i}} + 2\alpha \sum_{i\in\mathcal{Y},a\in\mathcal{A}} \frac{1}{\gamma_{i,a}\sqrt{n_{i,a}}}).$$

In the last formula, we multiple 2 for a nicer looking expression, it won't affect the optimal ratio for $\gamma_0/\gamma_1$. Also, notice that $\gamma_{i,a} \geq \gamma_i$, we have

$$\sqrt{C(\mathcal{F})}(\sum_{i\in\mathcal{Y}} \frac{1}{\gamma_i\sqrt{n_i}} + 2\alpha \sum_{i\in\mathcal{Y},a\in\mathcal{A}} \frac{1}{\gamma_{i,a}\sqrt{n_{i,a}}}) \leq \sqrt{C(\mathcal{F})}(\sum_{i\in\mathcal{Y}} \frac{1}{\gamma_i\sqrt{n_i}} + 2\alpha \sum_{i\in\mathcal{Y},a\in\mathcal{A}} \frac{1}{\gamma_i\sqrt{n_{i,a}}}).$$

The proof is complete.

### A.4.2 OPTIMIZATION OF $\gamma_0/\gamma_1$

**Theorem A.2** *For binary classification, let $\mathcal{F}$ be a class of neural networks with a bias term, i.e. $\mathcal{F} = \{f + b\}$ where $f$ is a neural net function and $b \in \mathbb{R}^2$ is a bas, with Rademacher complexity upper bound $\hat{\mathcal{R}}_*(\mathcal{F}) \leq \sqrt{\frac{C(\mathcal{F})}{n_*}}$. Suppose some classifier $f \in \mathcal{F}$ can achieve a total sum of margins $\gamma_0' + \gamma_1' = \beta$ with $\gamma_0', \gamma_1' > 0$. Then, there exists a classifier $f^* \in \mathcal{F}$ with margin ratio*

$$\gamma_0^*/\gamma_1^* = \tilde{n}_0^{-1/4}/\tilde{n}_1^{-1/4} = \tilde{n}_1^{1/4}/\tilde{n}_0^{1/4},$$

*where the adjusted sample size $\tilde{n}_i = \frac{n_i \Pi_a n_{i,a}}{(\sqrt{\Pi_a n_{i,a}} + \alpha \sum_{j\in\mathcal{A}} \sqrt{n_i \Pi_{a\in\mathcal{A}\backslash j} n_{i,a}})^2}$ for $i \in \mathcal{Y}$.*

*Proof:* This can directly follow the proof in Theorem 3 in Cao et al. (2019). The only difference is that we need to solve

$$\min_{\gamma_0+\gamma_1=\beta} \sum_{i\in\mathcal{Y}} \frac{1}{\gamma_i}\sqrt{\frac{C(\mathcal{F})}{n_i}} + \sum_{i\in\mathcal{Y},a\in\mathcal{A}} \frac{2\alpha}{\gamma_i}\sqrt{\frac{C(\mathcal{F})}{n_{i,a}}}.$$

More specifically, by simple calculation, for $\mathcal{Y} = \{0, 1\}$, $\mathcal{A} = \{a_1, a_2\}$,

$$\sum_{i\in\mathcal{Y}} \frac{1}{\gamma_i}\sqrt{\frac{1}{n_i}} + \sum_{i\in\mathcal{Y},a\in\mathcal{A}} \frac{2\alpha}{\gamma_i}\sqrt{\frac{1}{n_{i,a}}} = \frac{1}{\gamma_0}\sqrt{\frac{C(\mathcal{F})}{\tilde{n}_0}} + \frac{1}{\gamma_1}\sqrt{\frac{C(\mathcal{F})}{\tilde{n}_1}},$$

then by applying Theorem 3 in Cao et al. (2019) by replacing $n_i$'s with $\tilde{n}_i$'s, it gives the final result.

$\square$

### A.4.3 PROOF OF EXAMPLE 3.1.

**Example A.1 (Restatement of Example 3.1)** *Given function $f$ and set $S$, let*

$$\text{dist}(f, S) = \min_{x, s \in S} \|f(x) - s\|_2.$$

*Consider two classifiers $\tilde{f}, f \in \mathcal{F}$ such that*

$$\text{dist}(\tilde{f}, S_0) / \text{dist}(\tilde{f}, S_1) = \tilde{n}_0^{-1/4} / \tilde{n}_1^{-1/4}$$

*and $\text{dist}(f', S_0) / \text{dist}(f', S_1) = n_0^{-1/4} / n_1^{-1/4}$. Suppose $\|\beta^*\| \gg \sqrt{p \log n}$, $\|\mu_i\| < C$, $(\mu_1^* - \mu_2^*)^\top \beta = 0$, and $\pi_{1,a_2} \leq c_1 \pi_{1,a_1}$ for a sufficiently small $c_1 > 0$, then when $n_0, n_1$ are sufficiently large, with high probability we have $\mathcal{M}[\tilde{f}] < \mathcal{M}[f]$.*

*Proof:* Recall that $\tilde{n}_i = \frac{n_i \Pi_{a \in \mathcal{A}} n_{i,a}}{(\sqrt{\Pi_{a \in \mathcal{A}} n_{i,a}} + \alpha \sum_{a \in \mathcal{A}} \sqrt{n_i n_{i,a}})^2}$ for $i \in \{0, 1\}$, and our training data follow distribution: $x \mid y = 0 \sim \sum_{i=1}^{2} \pi_{0,a_i} \mathcal{N}_p(\mu_i, I)$, $x \mid y = 1 \sim \sum_{i=1}^{2} \pi_{1,a_i} \mathcal{N}_p(\mu_i + \beta^*, I)$.

$$
\begin{aligned}
\mathcal{M}[f] = &\frac{1}{2} \mathbb{P}(h(X) = 1 | Y = 0) + \frac{1}{2} \mathbb{P}(h(X) = 0 | Y = 1) \\
&+ \alpha \sum_{i \in \mathcal{Y}} |\mathbb{P}(h(X) = i | Y = i, A = a_1) - \mathbb{P}(h(X) = i | Y = i, A = a_2)| \\
= &\frac{1}{2} \sum_{i=1}^{2} \pi_{0,a_i} \Phi(\frac{\beta^{*\top} \mu_i - c}{\|\beta^*\|}) + \frac{1}{2} \sum_{i=1}^{2} \pi_{1,a_i} \Phi(\frac{c - \beta^{*\top} \mu_i - \|\beta^*\|^2}{\|\beta^*\|}) \\
&+ \alpha \cdot |\Phi(\frac{\beta^{*\top} \mu_0 - c}{\|\beta^*\|}) - \Phi(\frac{\beta^{*\top} \mu_1 - c}{\|\beta^*\|})| + \alpha \cdot |\Phi(\frac{\beta^{*\top} \mu_0 + \|\beta^*\|^2 - c}{\|\beta^*\|}) - \Phi(\frac{\beta^{*\top} \mu_1 + \|\beta^*\|^2 - c}{\|\beta^*\|})|
\end{aligned}
$$

For different margin ratio $\gamma$, we have $c = \mu_1^\top \beta^* + \frac{1}{1+\gamma} \|\beta^*\|^2 + O_P(\sqrt{p \log n})$, where the $O_P(\sqrt{p \log n})$ term accounts for the variation of random samples and is based on the fact that $\|Z\|^2 \sim \chi_p^2$ if $Z \sim \mathcal{N}_p(0, I)$ and $\max Z_i = O_P(p \log n)$ if $Z_1, ..., Z_n \overset{i.i.d.}{\sim} \chi_p^2$.

Using the fact that $\|\beta^*\| \gg \sqrt{p \log n}$ and $\|\mu_i\| < C$, we then have $c = (\frac{1}{1+\gamma} + o_P(1)) \|\beta^*\|^2$.

Similarly, we have

$$\mathcal{M}[f] = \Phi(-\frac{c}{\|\beta^*\|}) + \Phi(\frac{c - \|\beta^*\|^2}{\|\beta^*\|}) + o(1)$$

Then let's consider the two ratios $\gamma = n_0^{-1/4} / n_1^{-1/4}$ and $\tilde{\gamma} = \tilde{n}_0^{-1/4} / \tilde{n}_1^{-1/4}$.

In the following we compute $\tilde{n}_0$ and $\tilde{n}_1$:

When $\pi_{1,a_2} \leq c_1 \pi_{1,a_1}$ $\tilde{n}_1 = \frac{n_1 \Pi_{a \in \mathcal{A}} n_{1,a}}{(\sqrt{\Pi_{a \in \mathcal{A}} n_{1,a}} + \alpha \sum_{a \in \mathcal{A}} \sqrt{n_1 n_{1,a}})^2} \in (0.9 n_1, n_1)$

When $\pi_{0,a_2} = \pi_{0,a_1}$ $\tilde{n}_0 = \frac{n_0 \Pi_{a \in \mathcal{A}} n_{0,a}}{(\sqrt{\Pi_{a \in \mathcal{A}} n_{0,a}} + \alpha \sum_{a \in \mathcal{A}} \sqrt{n_0 n_{0,a}})^2} = (\frac{1}{1 + 2\sqrt{2}})^2 n_0$.

As a result, we have $\frac{\tilde{\gamma}}{\gamma} \in [1.9, 2]$. When the data is imbalanced such that $\gamma = (\frac{n_1}{n_0})^{1/4} > 1$, we have $0 < \frac{1}{1+\tilde{\gamma}} < \frac{1}{1+\gamma} < 1/2$, and consequently

$$\Phi(-\frac{\frac{1}{1+\tilde{\gamma}} \|\beta^*\|^2}{\|\beta^*\|}) + \Phi(\frac{\frac{1}{1+\tilde{\gamma}} \|\beta^*\|^2 - \|\beta^*\|^2}{\|\beta^*\|}) < \Phi(-\frac{\frac{1}{1+\gamma} \|\beta^*\|^2}{\|\beta^*\|}) + \Phi(\frac{\frac{1}{1+\gamma} \|\beta^*\|^2 - \|\beta^*\|^2}{\|\beta^*\|}).$$

Additionally, we have that when $\|\beta^*\| \to \infty$, the second term in the $\mathcal{M}[f]$, the fairness violation error is $o(1)$. Combining all the pieces, we have $\mathcal{M}[\tilde{f}] < \mathcal{M}[f]$.

$\square$

### A.4.4 PROOF OF THEOREM 5.1.

**Theorem A.3 (Restatement of Theorem 5.1)** *Let $\rho = \max_f \|M\hat{\mu}^{new}(f) - \hat{c}\|_\infty$. For $\eta = \nu/(2\rho^2 B)$, the modified **ExpGrad** will return a $\nu$-approximate saddle point of $L^{new}$ in at most $4\rho^2 B^2 \log(|\mathcal{K}| + 1)/\nu^2$ iterations.*

*Proof:* We consider an extended version of $h$ in Agarwal et al. (2018), which is a function of (x,y,a) instead of just be a function of $x$. $h : (x, y, a) \mapsto \{0, 1\}$. Notice that $\mu(h)$ also satisfies the requirement in Agarwal et al. (2018) with the extend version $h$. Thus, directly by classic result of in Freund & Scapire (1996) and theorm 1 in Agarwal et al. (2018), the result follows. □

### A.5 COMBINATION WITH OTHER ALGORITHMS

As we stated, the algorithm stated in the main context is just one of the examples that can be combined with our approach. FIFA can also be applied to many other popular algorithms such as fair representation (Madras et al., 2018). We here show how to combine with fair representation.

In Madras et al. (2018), there are several parts, an encoder $\rho$, an adversary $v$, a decoder $k$ and a predictor $g$. The optimization is:

$$\min_{g,\rho,k} \max_v \mathbb{E}_{X,Y,A} L(g, \rho, k, v),$$

where

$$L(g, \rho, k, v) = \lambda_1 \ell_c(g(\rho(X)), Y) + \lambda_2 \ell_{\text{dec}}(k(\rho(X), A), X) + \lambda_3 \ell_{\text{adv}}(v(\rho(X), A)),$$

for cross entropy loss $\ell_c$, decoding loss $\ell_{\text{dec}}$, and adversary loss $\ell_{\text{adv}}$. We can modify the cross entropy loss to $\ell_c$ to $\ell_{\text{FIFA}}$. So,

$$L_{\text{FIFA}}(g, \rho, k, v) = \lambda_1 \ell_{\text{FIFA}}(g(\rho(X)), Y) + \lambda_2 \ell_{\text{dec}}(k(\rho(X), A), X) + \lambda_3 \ell_{\text{adv}}(v(\rho(X), A)).$$

Actually, for $\ell_{\text{adv}}$, we can similarly modify for indices, but it is a little complicated and notation heavy, so we omit it here.

## B IMPLEMENTATION DETAILS

We use the official train-test split for the CelebA dataset. For AdultIncome and DutchConsensus, we use the `train_test_split` procedure of the `scikit-learn` package with training-test set ratio of $0.8$ and random seed of $1$ to generate the training and test set. We tabulate the sizes for subgroups in Table 3.

| Data | AdultIncome | | | | CelebA | | | |
|---|---|---|---|---|---|---|---|---|
| Label | — | | + | | — | | + | |
| Gender | Female | Male | Female | Male | Female | Male | Female | Male |
| Train | 9592 | 15128 | 1179 | 6662 | 71629 | 66874 | 22880 | 1387 |
| Test | 4831 | 7604 | 590 | 3256 | 9767 | 7535 | 2480 | 180 |

**Table 3:** Training and testing sample sizes for CelebA and AdultIncome datasets across labels (Label) and sensitive attributes (Gender).

The sweeps are done on the `wandb` platform (Biewald, 2020), where all hyper-parameters except for the grid, are searched using its built-in Bayesian backend. All models for the same dataset are trained with a fixed number of epochs where the training accuracies converge. Batch training with size 128 is used for CelebA and full batch training is used for AdultIncome. As a special case of FIFA, when $\delta_{i,a} = 0$ for all $i, a$ and $\alpha = 0$ the FIFA loss degenerates to non fairness-aware LDAM loss proposed in Cao et al. (2019); FIFA further finetunes $\delta_{i,a}$ and $\alpha$, and to ensure a fair comparison, we set the same coverage for the the common hyper-parameter $C$ in the sweeps. In Fig. 6, we show the histogram of the hyper-parameter $C$ in the sweeps for FIFA and LDAM. Note that the sweeps cover approximately the same range for this common hyper-parameter.

**Details on CelebA.** We use the same pre-processing steps as in (Sagawa et al., 2020) to crop the images in CelebA into $224 \times 224 \times 3$ and perform the same $z$-normalization for both training and test set. We use ResNet-18 models for training with the last layer being replaced to the `NormLinear` layer used by Cao et al. (2019) that ensures the input as well as the columns of the wight matrix (with 2 rows corresponding to each label class) has norm 1. This ensures our adjustments on the logits are comparable. We use the Adam optimizer with learning rate $1 \times 10^{-4}$ and weight decay $5 \times 10^{-5}$ to train these models with stochastic batches of sizes 128. We performed pilot experiments and learnt

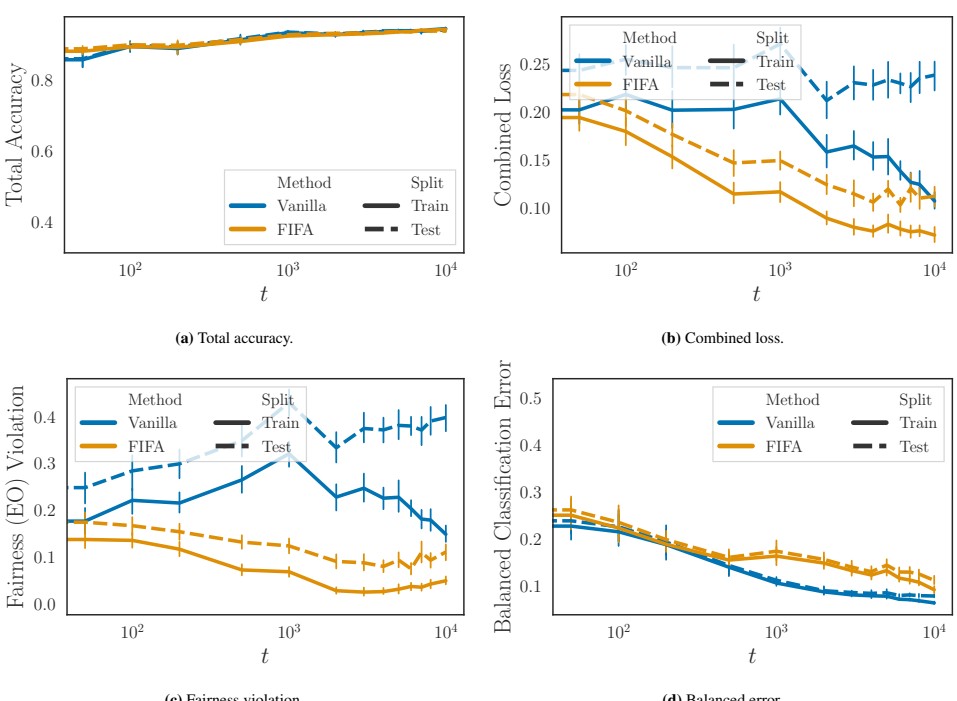

**(a)** Total accuracy.

**(b)** Combined loss.

**(c)** Fairness violation.

**(d)** Balanced error.

**Figure 7:** Training and test trajectories of different metrics of ResNet-18 on CelebA dataset under FIFA and vanilla losses respectively. We note that the generalization performance of vanilla models are consistently poor as training time increases, suggesting that it is difficult to cultivate an early-stopping scheme that might alleviate poor fairness generalization.

that under this configuration the models usually converges within the first 1500 iterations in terms of training losses and thus we fix the training time as 8000 iterations which corresponds to roughly four epochs.

**AdultIncome and DutchConsensus.** AdultIncome and Dutch-Consensus are two relatively smaller datasets that have been used for benchmarking for various fair classification algorithms such as Agarwal et al. (2018). We convert all categorical variables to dummies and use the standard $z$-normalization to pre-process the data. There are 107 features in AdultIncome and 59 in DutchConsensus, both counting the sensitive attribute, gender. We intend to use these datasets to test smaller models such as logistic regression, and we implement it as a one-layer neural net for consistency concerns, which is trained using full-batch gradient descent using Adam with learning

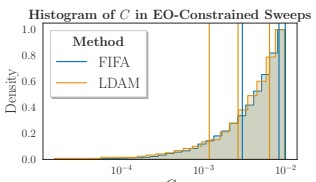

**Figure 6:** Histogram (cumulative density) of hyper-parameter $C$ in the sweeps for FIFA and LDAM. Vertical lines mark the values corresponding to the best performing models in Table 1.

rate $1 \times 10^{-4}$ and weight decay $5 \times 10^{-5}$ for 10000 epochs. We set $\delta_{0,\text{Female}}, \delta_{1,\text{Female}} \in [0, 0.01]$, and $\delta_{0,\text{Male}} = \delta_{1,\text{Male}} = 0$ for the AdultIncome dataset and $\delta_{0,\text{Male}}, \delta_{1,\text{Female}} \in [0, 0.01]$, and $\delta_{0,\text{Female}} = \delta_{1,\text{Male}} = 0$ for the DutchConsensus Dataset. All models converge after this training measured by the training metrics. Although the exact pre-processing procedures for these two datasets are not available in Agarwal et al. (2018), we found that on vanilla models under both GridSearch and ExponentiatedGradient methods, the training and test performance (measured by total accuracy and fairness violation) are comparable with those reported in Agarwal et al. (2018).

**Computational resource considerations.** We perform all experiments on NVIDIA GPUs RTX 2080 Ti. Each experiment on CelebA usually takes less than two hours (clocktime) and each experiment on AdultIncome and DutchConsensus takes less than ten minutes.

# C  ADDITIONAL EXPERIMENTAL RESULTS

## C.1  A TRAJECTORY ANALYSIS ON CELEBA

One observation we made in Table 1 is that the improvements of the generalization of combined loss on CelebA is largely due to the improved generalization performance on fairness violations. It is natural to wonder whether this behavior suggests that the sweet spots of generalization performance for balanced error and fairness violation may not be aligned, i.e., there is a difference in training time scales for these two metrics to reach their optimal generalization. Furthermore, it is also open that whether one could enforce certain early stopping procedure (e.g., on the combined loss or the fairness violation) such that the generalizations on vanilla models may be improved.

To explore these two questions, we plot the trajectories of training and test metrics for FIFA and vanilla (hyper-parameter chosen to be those corresponding to the best-performing models in Table 1) in Fig. 7. We observe that it is difficulty to (i) identify sweet spots of generalization gaps for the vanilla models; and (ii) enforce a reasonable early stopping criterion that improves the generalization performances thereof.

## C.2  CELEBA AND THE DP CONSTRAINT

We presented in Table 1 our main results, CelebA dataset trained with grid search under EO constraint. We show in Table 4 the results on the DP constraints. Here all training configurations are the same as Table 1, except that we replace the EO constraint by the DP constraint. For ease of comparison, we also recall the results on EO in Table 4. The observations are similar to those we made for Table 1, namely, FIFA improves significantly on the combined loss compared with vanilla.

| | | EO | | | | DP | | | |
|---|---|---|---|---|---|---|---|---|---|
| | | **Combined Loss** | | **Fairness Violation** | | **Combined Loss** | | **Fairness Violation** | |
| $\epsilon$ | **Method** | Train | Test | Train | Test | Train | Test | Train | Test |
| 0.01 | FIFA | 7.37% | 6.71% | 5.31% | 2.75% | 8.65% | 7.21% | 4.84% | 1.45% |
| | Vanilla | 7.14% | 14.01% | 6.69% | 20.29% | 10.74% | 10.43% | 4.72% | 1.35% |
| 0.05 | FIFA | 5.46% | 6.34% | 2.63% | 3.29% | 10.02% | 9.40% | 4.73% | 1.07% |
| | Vanilla | 8.84% | 13.05% | 9.45% | 17.92% | 12.14% | 11.60% | 8.34% | 5.17% |
| 0.10 | FIFA | 5.92% | 6.54% | 3.11% | 2.65% | 9.04% | 8.32% | 2.98% | 0.04% |
| | Vanilla | 8.90% | 16.71% | 11.37% | 26.15% | 12.13% | 11.66% | 9.77% | 6.83% |

**Table 4:** Grid search with EO and DP constraint on CelebA dataset (Liu et al., 2015) using ResNet-18, best results with respect to test combined loss among sweeps of hyper-parameters are shown.

## C.3  GRID SEARCH ON ADULTINCOME (DP AND EO)

| | | EO | | | | DP | | | |
|---|---|---|---|---|---|---|---|---|---|
| | | **Combined Loss** | | **Fairness Violation** | | **Combined Loss** | | **Fairness Violation** | |
| $\epsilon$ | **Method** | Train | Test | Train | Test | Train | Test | Train | Test |
| 0.01 | FIFA | 14.77618% | 14.93573% | 8.53851% | 8.50086% | 13.75700% | 14.05881% | 0.08609% | 0.00999% |
| | Vanilla | 16.68724% | 17.28659% | 10.39000% | 10.92794% | 14.83347% | 15.09909% | 3.32436% | 3.65903% |
| 0.05 | FIFA | 14.79263% | 14.91599% | 8.57436% | 8.50841% | 13.74952% | 14.03440% | 0.11811% | 0.01008% |
| | Vanilla | 16.68724% | 17.28659% | 10.39000% | 10.92794% | 14.83475% | 15.09909% | 3.32895% | 3.65903% |
| 0.10 | FIFA | 14.70959% | 14.88935% | 8.17597% | 8.16283% | 13.72278% | 14.03193% | 0.11331% | 0.01011% |
| | Vanilla | 16.68724% | 17.28659% | 10.39000% | 10.92794% | 14.82782% | 15.09909% | 3.31507% | 3.65903% |

**Table 5:** Grid search with EO and DP constraint on AdultIncome dataset using logistic regression, best results with respect to test combined loss among sweeps of hyper-parameters are shown.

We also give in Table 5 the grid search results on AdultIncome dataset, for both EO and DP constraints. We observe that on this small dataset and small model, FIFA can also improve generalization performances for both EO and DP constraints. This further exhibits the flexibility of the FIFA approach.

| | | Combined Loss | | Fairness Violation | | Balanced Error | |
|---|---|---|---|---|---|---|---|
| | | Train | Test | Train | Test | Train | Test |
| $\epsilon$ | Method | | | | | | |
| 0.01 | FIFA | 14.26% | 13.60% | 5.39% | 4.53% | 23.12% | 22.67% |
| | Vanilla | 14.22% | 13.68% | 5.33% | 4.69% | 23.10% | 22.67% |
| 0.05 | FIFA | 14.02% | 13.22% | 4.93% | 3.71% | 23.11% | 22.73% |
| | Vanilla | 14.00% | 13.23% | 4.89% | 3.71% | 23.11% | 22.74% |
| 0.10 | FIFA | 14.20% | 13.67% | 5.29% | 4.67% | 23.10% | 22.67% |
| | Vanilla | 14.20% | 13.68% | 5.30% | 4.67% | 23.10% | 22.68% |

**Table 6:** FIFA+ExpGrad on the New Adult Income dataset.

## C.4 EXPGRAD ON THE NEW ADULTINCOME DATASET

We give in Table 6 the results by applying FIFA+ExpGrad on the new adult income dataset. We use the employment data from California in 2021 and set the same threshold of yearly income being 50K to construct the label. However, the sensitive attributes becomes the race, where 0 for White and 1 for Black. We use the recommended covariates, which after dummification, results in 20 covariates and 129563 samples. We split the samples randomly into 80% training set and 20% test set. The other configurations remain the same.

## C.5 PER-GROUP RESULTS

To have a better understanding of the per-group performance, we take a deeper look at (i) CelebA using FIFA+GridS; (ii) AdultIncome using FIFA+GridS; and (iii) New AdultIncome using FIFA+ExpGrad, all optimizing over the EO constraint. We show in Table 7 per-group accuracies and in Table 10 the differences between FPR and FNR across sensitive groups. Note that since the New AdultIncome only conatins 20 covariates, the performances of different methods differ insignificantly.

| | | Non-Blond Female | | Non-Blond Male | | Blond Female | | Blond Male | |
|---|---|---|---|---|---|---|---|---|---|
| | | Train | Test | Train | Test | Train | Test | Train | Test |
| $\epsilon$ | Method | | | | | | | | |
| 0.01 | FIFA-GS | 95.28% | 95.50% | 93.01% | 92.70% | 88.20% | 87.22% | 92.86% | 84.44% |
| | LDAM-GS | 95.37% | 95.47% | 94.21% | 94.19% | 88.81% | 87.06% | 91.13% | 81.67% |
| | Vanilla-GS | 89.15% | 90.29% | 95.85% | 96.12% | 92.78% | 93.06% | 86.88% | 72.78% |
| 0.05 | FIFA-GS | 95.13% | 95.11% | 93.50% | 93.19% | 88.93% | 87.18% | 91.56% | 83.89% |
| | LDAM-GS | 94.97% | 94.90% | 91.83% | 92.01% | 87.15% | 86.29% | 92.86% | 82.22% |
| | Vanilla-GS | 86.15% | 87.50% | 94.72% | 95.16% | 93.81% | 94.03% | 84.35% | 76.11% |
| 0.10 | FIFA-GS | 95.61% | 95.75% | 93.15% | 93.10% | 87.95% | 84.68% | 91.06% | 82.78% |
| | LDAM-GS | 96.14% | 96.15% | 93.69% | 93.19% | 84.57% | 81.85% | 91.49% | 79.44% |
| | Vanilla-GS | 87.07% | 88.03% | 97.75% | 97.56% | 95.58% | 95.04% | 84.21% | 68.89% |

**Table 7:** Per-group accuracy on the CelebA dataset from FIFA+GridS (EO).

| | | <50K Female | | <50K Male | | >=50K Female | | >=50K Male | |
|---|---|---|---|---|---|---|---|---|---|
| | | Train | Test | Train | Test | Train | Test | Train | Test |
| $\epsilon$ | Method | | | | | | | | |
| 0.01 | FIFA-GS | 95.30% | 95.22% | 86.76% | 86.72% | 65.73% | 63.56% | 68.28% | 67.91% |
| | Vanilla-GS | 74.64% | 74.39% | 64.25% | 63.47% | 89.91% | 88.31% | 85.02% | 84.40% |
| 0.05 | FIFA-GS | 95.29% | 95.24% | 86.71% | 86.73% | 65.65% | 63.39% | 68.34% | 68.03% |
| | Vanilla-GS | 74.64% | 74.39% | 64.25% | 63.47% | 89.91% | 88.31% | 85.02% | 84.40% |
| 0.10 | FIFA-GS | 95.32% | 95.30% | 87.14% | 87.14% | 65.22% | 63.22% | 67.55% | 67.05% |
| | Vanilla-GS | 74.64% | 74.39% | 64.25% | 63.47% | 89.91% | 88.31% | 85.02% | 84.40% |

**Table 8:** Per-group accuracy on the AdultIncome dataset from FIFA+GridS (EO).

| $\epsilon$ | Method | <50K White | | <50K Black | | >=50K White | | >=50K Black | |
|---|---|---|---|---|---|---|---|---|---|
| | | Train | Test | Train | Test | Train | Test | Train | Test |
| 0.01 | FIFA-GS | 71.34% | 71.96% | 70.03% | 72.40% | 82.79% | 82.92% | 77.40% | 78.38% |
| | Vanilla-GS | 71.41% | 71.99% | 69.96% | 72.22% | 82.78% | 82.91% | 77.44% | 78.22% |
| 0.05 | FIFA-GS | 71.53% | 72.01% | 69.37% | 71.68% | 82.67% | 82.76% | 77.74% | 79.04% |
| | Vanilla-GS | 71.54% | 72.00% | 69.30% | 71.58% | 82.67% | 82.76% | 77.78% | 79.04% |
| 0.10 | FIFA-GS | 71.40% | 72.00% | 69.96% | 72.22% | 82.78% | 82.89% | 77.48% | 78.22% |
| | Vanilla-GS | 71.40% | 71.98% | 69.96% | 72.22% | 82.78% | 82.89% | 77.48% | 78.22% |

**Table 9:** Per-group accuracy on the New AdultIncome dataset from FIFA+ExpGrad (EO).

| $\epsilon$ | Method | CelebA (GridS) | | | | AdultIncome (GridS) | | | | New AdultIncome (ExpGrad) | | | |
|---|---|---|---|---|---|---|---|---|---|---|---|---|---|
| | | FPR | | FNR | | FPR | | FNR | | FPR | | FNR | |
| | | Train | Test | Train | Test | Train | Test | Train | Test | Train | Test | Train | Test |
| 0.01 | FIFA | 2.27% | 2.79% | 4.66% | 2.77% | 8.54% | 8.50% | 2.55% | 4.35% | 1.31% | 0.44% | 5.39% | 4.53% |
| | Vanilla | 6.69% | 5.83% | 5.90% | 20.29% | 10.39% | 10.93% | 4.89% | 3.91% | 1.45% | 0.23% | 5.33% | 4.69% |
| 0.05 | FIFA | 1.63% | 1.91% | 2.63% | 3.29% | 8.57% | 8.51% | 2.69% | 4.64% | 2.17% | 0.33% | 4.93% | 3.71% |
| | Vanilla | 8.57% | 7.66% | 9.45% | 17.92% | 10.39% | 10.93% | 4.89% | 3.91% | 2.24% | 0.41% | 4.89% | 3.71% |
| 0.10 | FIFA | 2.46% | 2.65% | 3.11% | 1.90% | 8.18% | 8.16% | 2.32% | 3.83% | 1.44% | 0.22% | 5.29% | 4.67% |
| | Vanilla | 10.68% | 9.53% | 11.37% | 26.15% | 10.39% | 10.93% | 4.89% | 3.91% | 1.44% | 0.24% | 5.30% | 4.67% |

**Table 10:** FPR/FNR differences across sensitivy groups on three datasets.

