# OpenReview forum: "FIFA: Making Fairness More Generalizable in Classifiers Trained on Imbalanced Data"
_ICLR.cc/2023/Conference — ICLR 2023 poster_

### Official Review · Reviewer_WVpp · 2022-10-23

**Confidence:** 3
**Correctness:** 4
**Technical Novelty And Significance:** 2
**Empirical Novelty And Significance:** 3
**Recommendation:** 6

**Clarity, Quality, Novelty And Reproducibility:**

The work is clear, reasonably easy to follow and seems novel to the best of my knowledge. The reviewer would be able to reproduce these based on the details provided.

**Details Of Ethics Concerns:**

None, so far. The authors have addressed this in pg 10.

**Strength And Weaknesses:**

Strengths
* The paper is well written, clear to follow.
* It motivates the problem and lists the related work reasonably.It tackles an important problem in the domain, especially with the advent of large  overparameterized models.


Weakness
* The novelty of the technical contributions itself doesn't seem quite significant, specifically the integration with existing classification based approaches.
* The overall empirical evaluation is  reasonably extensive, answering the questions raised in the introduction. However, it would be nice if the authors could elucidate further the  complexity/real world mapping of these datasets to real world challenges we observe.

**Summary Of The Paper:**

The work studies the generalization of fairness constraints  in Machine Learning systems.  The authors argue that while current algorithms are optimized for generalizing on the whole dataset, their efficacy on minority classes for imbalanced datasets have not been extensively covered They introduce their Flexible Imbalance Fairness Aware (FIFA) classification, combined with logit based losses.  The authors conclude their work by empirically showcasing the performance on relevant datasets for over-parameterized models.

**Summary Of The Review:**

Overall, the paper tackles an important problem in the domain, especially with the rapid growth of large overparameterized. The paper is clear, well written and easy to follow. While the empirical significance is reasonably novel, the introduced technical novelty isn't as significant.

---

> ### Author Response · Authors · 2022-11-16
> **Response to Reviewer WVpp**
>
> Thanks for your positive feedback! Here, we provide answers to your concerns point by point.
>
> **Q1:** The novelty of the technical contributions itself doesn't seem quite significant, specifically the integration with existing classification based approaches.
>
> **A1:** Thanks for your comment. We want to emphasize that the flexibility to integrate with existing methods is actually a main advantage of our method. Section 5 is just *an example*. Our method can be combined with any logit-based fair classification methods to improve their performance. Also, we introduce some new techniques to deal with $\delta_{i,a}$ based on re-arrangement inequality. Meanwhile, we theoretically provide an example in Example 3.1 to demonstrate the advantage of our method compared to Cao et. at. 2019.
>
>
> **Q2:** ...it would be nice if the authors could elucidate further the complexity/real world mapping of these datasets to real world challenges we observe.
>
>
> **A2:** Thanks for your comment. In Figure 2 in our paper, we plot the ratio of each subgroup in real data sets. We can see that the subgroups (sensitive attribute, either Male or Female, and label class, either + or
> -) are very imbalanced in many popular datasets
> across different domains.
> In fact, during the rebuttal period, we also looked into
> the new Adult Income data by Ding et al. suggested by Review Kgun (specifically, we look at the data from California in 2021).
> Note that this data is more temporally
> and spatially relevant to real-world; yet
> we still observe the same challenges
> in the original datasets, namely, imbalance between subgroups, as shown below:
>
> |       | >=50K |       | <50K  |       |
> |-------|-------|-------|-------|-------|
> |       | White | Black | White | Black |
> | Train | 42996 | 2345  | 53785 | 4508  |
> | Test  | 10654 | 606   | 13571 | 1098  |
>
> Thus we think the difficulties introduced by the imbalance nature of real-world data are still relevant and prevalent. Moreover, our experiments on some of the standard datasets such as CelebA and various Adult Income datasets complement our theory and support the benefit of our FIFA method to practitioners.
>
> While we mainly focused on the dataset imbalance in this work -- as we feel it the most preeminent and under-studied -- it only captures one side of a kaleidoscope of complexities in real-world. We are also interested in further studying how other artefacts such as selection bias and spurious correlations, say, may come into play as far as fair classifiers are concerned. These are interesting directions for future work.

---

### Official Review · Reviewer_Kgun · 2022-10-26

**Confidence:** 4
**Correctness:** 3
**Technical Novelty And Significance:** 3
**Empirical Novelty And Significance:** 3
**Recommendation:** 6

**Clarity, Quality, Novelty And Reproducibility:**

The paper is well-written and mostly free of typos. While certain parts of the work were dense, at a high level the work was easy to follow.

The key insight of this work is mostly original even though significant portions of the work build on the previous Cao et. al paper as well as the formulation of Agarwal et. al.

Overall, the work addresses an important problem with a new margin-based loss function that improves the generalization of the fairness property of a model at test-time.

**Strength And Weaknesses:**

**Strengths**\
The paper addresses an important problem and the choice of the loss does indeed make intuition sense. Further, the empirical results indeed suggest that one makes significant improvements using the proposed loss function modifications in this work.

**Weaknesses**
- Clarification about the importance of group sizes: I am confused about why the bounds in the paper and the key intuition is around group size as opposed to some other measure of group difficulty. One could envision a case where a smaller group is perhaps more homogenous, so it still easy to 'learn' with smaller samples versus a larger group whose inputs have more feature noise. I would've expected a term in these bounds that captures group difficulty. Any insight on this issue?
- Adult dataset: I'd suggest the authors consider the recently introduced adult dataset by Ding et. al. (Retiring Adult dataset) as a replacement for the standard Adult dataset.
- On first reading, it was unclear what parts of section 5 are new to this work, and which ones aren't. For example, I know the ExpGrad formulation is due to Agarwal, but several parts of that section are not well-delineated. For example, is Theorem 5.1 also based on the modification of the equivalent theorem from the Agarwal paper?



**Summary Of The Paper:**

In settings where the goal is to learn a classifier that does not exhibit disparate impact, it is common to constrain the learning process via fairness constraints or an adversarial learning procedure. Typically, the disparate impact effects are reduced on the training set, but bias is still observed on the test set. In this paper, the authors propose a new margin-based loss function that is then minimized via exponentiated gradient descent. The intuition behind the loss is that the decision boundary of the final classifier should be further away from small sub-groups in the dataset.

**Summary Of The Review:**

Overall, the work addresses an important problem with a new margin-based loss function that improves the generalization of the fairness property of a model at test-time.

---

> ### Author Response · Authors · 2022-11-16
> **Response to Reviewer Kgun**
>
> Thanks for your insightful and positive feedback! Here, we provide answers to your concerns point by point.
>
> **Q1:** Clarification about the importance of group sizes...
>
>
> **A1:** Thanks for your insightful question! Our key intuition for group size is because we are doing *distribution-free* analysis on the generalization behavior of the classifier, i.e., we do not assume any prior knowledge about the data distribution. Our theoretical bounds on $\mathcal{M}[f]$ is mainly in the flavor of uniform generalization bounds, e.g. classic Rademacher complexity bounds. From our results, we can see that our method encourages large margin for the minority group, because  when the sample size is small, what we learn from the small samples may not accurately capture the underlying true distribution of that group. In this case, a large margin  helps because it makes the classifier more tolerant to outliers. If we allow prior knowledge of data distributions, then indeed the generalization behavior of the classifier is related to the group difficulty. In an extreme case, if one group's distribution is degenerated --- all probability mass is concentrated on one point, then there will be no generalization issue for that group --- one point is enough to capture the true distribution, which makes designing margin for that group extremely easy. In contrast, our framework is aiming for more general case, where we have no prior knowledge of the data distribution.
>
>
>
>
>
>
> **Q2:** ... I'd suggest the authors consider the recently introduced adult dataset by Ding et. al. (Retiring Adult dataset)...
>
>
> **A2:** Thanks for your suggestion. We add a result in the appendix in the revised paper for the retiring adult dataset. Specifically, Table~6 provides new results
> by applying FIFA+ExpGrad on the new adult income dataset.
> We use the employment data from California in 2021
> and set the same threshold of yearly income being $50$K
> to construct the label. However, the sensitive attributes
> becomes the race, where $0$ for White and $1$ for Black.
> We use the recommended covariates, which after dummification, results in $20$ covariates and $129563$
> samples. We split the samples randomly into $80\%$
> training set and $20\%$ test set. The other configurations remain the same.
>
> Meanwhile, we provide results in Table 9 and 10 for per-group results on the retiring adult dataset, i.e. per-group accuracy and differences between FPR and FNR across sensitive group. Please see the appendix in the revised PDF for details.
>
>
>
>
>
> **Q3:** ...it was unclear what parts of section 5 are new to this work, and which ones aren't...
>
> **A3:** Thanks for your question. The new part starts from the sentence ``To combine our approach", where we start describing how to combine our FIFA framework to Agarwal's work. We also demonstrate an algorithm in Algorithm 1, which is a new algorithm based on our method. For Theorem 5.1, it is a counterpart of  Theorem 1 in Agarwal et.al., which demonstrates that combining with FIFA, the modified algorithm enjoys similar good optimization theoretical guarantee as the original method, by using similar derivation techniques. We have updated Sec. 5 to highlight the components new to this work in red.

---

### Official Review · Reviewer_XQ8s · 2022-11-01

**Confidence:** 3
**Correctness:** 3
**Technical Novelty And Significance:** 2
**Empirical Novelty And Significance:** 2
**Recommendation:** 6

**Clarity, Quality, Novelty And Reproducibility:**

- Paper is well written and clearly presented.
- This work has clear close ties with other papers in the generalization literature, but I believe its application to fairness _with imbalanced data_ is novel enough to warrant a publication.

**Strength And Weaknesses:**

Strengths:
1. Margin-based approach makes sense in theory and by intuition.
2. Good comparisons against the well-known reductions-based approaches by Agarwal et al. (2018).
3. Good empirical results.

Weaknesses:
- Would be interesting to have comparisons against more than a single work; namely, a glaring baseline that's missing is the TensorFlow Constrained Optimization package, even more so given that this work focuses on over-parameterized models (NNs).
  - The same authors of the TFCO package also published work specifically focused on generalization of fairness constraints:
> Cotter, Gupta, Jiang, Srebro, Sridharan, Wang, Woodworth and You. "Training Well-Generalizing Classifiers for Fairness Metrics and Other Data-Dependent Constraints".
  - These references should at least be discussed in related work, but in truth they should also serve as baselines for empirical results.
- Bad fairness constraint generalization can only mean that performance on one subgroup generalizes better than performance on the other group; it would be interesting to see group-wise loss (or group-wise TPR or FPR), and how these change with difference values for fairness constraint tolerance.
  - Since this difference in generalization between different subgroups likely stems from the imbalanced nature of the dataset, it would be interesting to see comparison against methods from the imbalanced learning literature (not only the single method from fairness literature).

Minor point:
- Most references have double parentheses.

**Summary Of The Paper:**

The paper presents a regularization-based training loss modification that promotes larger margins for minority subgroups in order to improve fairness constraint generalization.

**Summary Of The Review:**

The paper presents a somewhat novel approach to improving fairness constraint generalization based on enlarging margins of minority subgroups. It is missing some discussion of important related work, and important baselines for fairness constraint generalization as well.

---

> ### Author Response · Authors · 2022-11-16
> **Response to Reviewer XQ8s (1/2)**
>
> Thank you for your thoughtful and positive feedback! Here, we provide answers to your suggestion point by point.
>
> **Q1:** Would be interesting to have comparisons against more than a single work...
>
>
> **A1:** Thanks for pointing out this seminal related work.
>
> (i). We have added discussion of the relationship between our work and [1] in the revised paper (colored red in Related Works):
>
> In [1], they investigate the generalization of constrained optimization with data-dependent constraints by framing the problem as a two-player zero-sum game. They provide an oracle-based algorithm in  and an algorithm without oracle but with strong assumptions including strong convexity. However, their practical algorithm for real-data is a heuristic one described in Section 5. Their results are shown to improve the generalization of constraints in linear models and one-hidden-layer NNs on both simulated data and real data (see Table 4 and Figure 2 in their paper). Unlike our work, they do not address the inherent imbalance in real datasets, and also their experimental results are not implemented with large NNs that people use in practice (we have results for Resnet-18, but the largest NN they use is a one-hidden-layer NN with 100 hidden units). *The most important difference* is that our work provides a flexible framework to combine with other fair learning methods. We demonstrate that in point (ii) below how our framework can even be combined with [1].
>
> (ii). In addition, we find our work can also be combined with [1]. Let us take the optimization with equalized opportunity in Appendix A.2 of [1] as an example. Specifically, the optimization problem is:
> $$
> \min _ {\theta\in\Theta}\mathbb{E} _ {x,y}[I \\{ y f(x;\theta) \le 0 \\} ],
> $$
> such that for all $i$,
> $$
> \mathbb{E} _ {x,y|y=1\wedge x_k=i}[I\\{yf(x;\theta)\le 0\\}]\le 1.1\cdot\mathbb{E} _ {x,y|y=1}[I\\{yf(x;\theta)\le 0\\}],
> $$
> where $I$ is the indicator and $x_k$ is the coordinate of $x$ that indicates which group the individual belongs to. Notice that
> $$
> I\\{yf(x;\theta)\le 0\\}=I\\{(g(x;\theta)) _ y\le \max _ {j\in \mathcal Y\backslash y} (g(x,\theta)) _ j\\}
> $$
> where $g(x,\theta)=(f(x,\theta),0)$. This is recovered by the 0-1 loss case discussed in our paper in the paragraph ``Flexible imbalance-fairness-aware (FIFA) approach" in Section 4. We can modify it by
> $$
> I\\{(g(x;\theta)) _ y-\Delta _ {y,i}\le \max _ {j\in \mathcal Y\backslash y} (g(x,\theta)) _ j\\}
> $$
> for $x$ with $x_k=i$, where $\Delta_{j,i}=C/\tilde n^{1/4} _ {j}+\delta _ {j,i}$. Also, by taking $C=0$ and $\delta _ {i,i}=0$ in the $\Delta$ in our paper, it is the original optimization.
>
> (iii). We looked into the code provided by [1]. Their code is written in TensorFlow and they implemented their methods on linear models/one-hidden-layer NNs for relatively small datasets. In contrast, we mainly use Pytorch and also need to deal with complex structures and large datasets, so it takes some time to setup a new baseline for all cases in our paper.
>
> Given the short period of rebuttal phase, we implement one case as a case study. We implemented their code on Adult dataset in our model settings for demographic parity (see details in C.3 in the appendix.) The combined loss (involving balanced error, not just error) for their method is $15.3\\%$ and $3.2\\%$ for the fairness violation, while our method by using FIFA+ExpGrad outperforms theirs in both combined losses value and fairness violation --- for instance taking  $\epsilon=0.05$, our combined loss is $14.0\\%$ and fairness violation is $0.01\\%$ (notice here the violation is true violation without any relaxation).

---

> ### Author Response · Authors · 2022-11-16
> **Response to Reviewer XQ8s (2/2)**
>
>
>
> **Q2:** ... it would be interesting to see group-wise loss (or group-wise TPR or FPR)...
>
>
> **A2:** Thanks for the suggestions. To address your suggestion, we added experiments in Section C.5 in the appendix in the revised PDF. We take a deeper look at
> (i) CelebA using FIFA+GridS; (ii) AdultIncome using FIFA+GridS; and (iii) New AdultIncome (Retiring Adult result in Ding et. al.) using FIFA+ExpGrad,
> all optimizing over the EO constraint.
> We show in Table 7
> per-group accuracies and in Table 10
> the differences between FPR and FNR across sensitive groups. Note that since the New AdultIncome only contains
> $20$ covariates, the performances of different methods
> differ insignificantly.
>
> In addition, we want to mention that comparison against methods from the imbalanced learning literature has been discussed extensively (without considering fairness constraints), for example see the related work part in Cao et. al. For literature about imbalanced datasets with fairness constraints, as far as we know, we are the first one to consider this problem.
>
>
> **Q3:** Most references have double parentheses.
>
> **A3:** We have fixed that in our revised version. Thanks for your careful examination!
>
> [1] Cotter, Gupta, Jiang, Srebro, Sridharan, Wang, Woodworth and You. ``Training Well-Generalizing Classifiers for Fairness Metrics and Other Data-Dependent Constraints".

---

### Author Response · Authors · 2022-11-16
**Summary of Revision**

We want to thank all three reviewers for reviewing our paper and for their constructive feedbacks. We have revised the draft accordingly, with major changes highlighted below:

1. Per Reviewer XQ8s's suggestion, we added detailed discussions on Cotter et al.'s work; we also studied a case and made comparisons between our results and theirs. We also **add per-group results** as suggested in Appendix C.5. For more details, please see the individual response.

2. Per Reviewer Kgun's suggestion, we looked into the new AdultIncome dataset introduced by Ding et al. (we used data from California in 2021 and follow the guidelines in their paper to preprocess the dataset). **We added four new tables in Appendix C,** showing results on the new AdultIncome, as well as per-group accuracies and FPR/FNR difference across groups.

---

### Decision · Program_Chairs · 2023-01-20

**Decision:**

Accept: poster

**Justification For Why Not Higher Score:**

Decent, but not groundbreaking technical novelty.

**Justification For Why Not Lower Score:**

Three positive reviews owing to a well-executed paper. From my reading, I concur with this view.

**Metareview: Summary, Strengths And Weaknesses:**

The paper proposes a new approach to ensure fairness on problems with label imbalance. The idea is to construct a flexible imbalance-fairness-aware (FIFA) loss, which builds on existing label-aware margin losses, to incorporate a margin dependent on the labels but also the sensitive attributes.

All three reviewers had a positive view of the paper, finding it a generally well-written, well-executed work that has clear intuitions and theoretical support, building on margin losses for label imbalance. One potential critique was that the final proposal is a somewhat natural combination of existing ideas (LDAM, and fairness reductions); nonetheless, the paper does take care to spell out the motivations and theoretical underpinnings for their final solution. Overall, the paper could be of interest to the community, and could help inspire more work in the area.

**Note From Pc:**

if the above contains the word "oral" or "spotlight" please see: "oral" presentation means -> notable-top-5% and "spotlight" means -> notable-top-25%. As stated in our emails, we are disassociating presentation type from AC recommendations